# A metabolomic signature of maternal BMI is associated with pregnancy complications across two independent pregnancy cohorts

David Horner [1] ✉, Rebecca Vinding[1], Tingting Wang [1], Mina Ali[1], Mario Lovric [1], Nicole Prince [2], Jessica Lasky-Su[2], Klaus Bønnelykke [1], Jakob Stokholm [1,3], Bo Chawes [1] & Morten Arendt Rasmussen [1,3]

## Abstract

**Background** Maternal obesity is increasingly common and linked to pregnancy complications, likely driven by underlying metabolic perturbations. This study investigates the association between maternal pre-pregnancy body mass index (BMI) and pregnancy complications through blood metabolomics, aiming to identify specific metabolites mediating these associations. **Methods** Data from the Copenhagen Prospective Studies on Asthma in Childhood 2010 (COPSAC2010) and Vitamin D Antenatal Asthma Reduction Trial (VDAART) cohorts were used, with untargeted blood metabolomics performed on blood samples taken during early-, mid-, and late gestation. Associations were assessed using multivariable logistic regression and mediation analyses to explore metabolite pathways linking maternal BMI with pregnancy complications. **Results** In the COPSAC2010 cohort, maternal pre-pregnancy BMI is associated with gestational diabetes (OR 1.90 [1.29-2.74], p = 6.75×10$^{-4}$), caesarean section (OR 1.23 [1.03-1.47], p = 0.023), and birth induction (OR 1.42 [1.21-1.67], p = 2.86×10$^{-5}$). A BMI-associated metabolite score is even more strongly associated with these complications and is independently associated with preeclampsia (OR 1.54 [1.04-2.26], p = 0.030). Validation in the VDAART cohort confirms the predictive value of the metabolite score for gestational diabetes (OR 2.10 [1.48-3.03], p = 4.97×10$^{-5}$) and preeclampsia (OR 2.12 [1.32-3.47], p = 0.002), particularly in late gestation. Mediation analysis in COPSAC2010 identifies 16 metabolites as mediating the effect of BMI on gestational diabetes. A model based on this subset of metabolites significantly outperforms the full maternal BMI model in predicting outcomes during both early (p = 0.009) and late gestation (p = 0.016) in the VDAART cohort. **Conclusions** These findings suggest that integrating metabolomic profiling into prenatal care could improve the prediction and management of adverse pregnancy outcomes.

## Plain Language Summary

Many women enter pregnancy with a higher body weight, which increases the risk of complications such as diabetes, preeclampsia, and delivery by caesarean section. We wanted to understand why these risks occur and whether changes in the body's metabolism could explain them. To investigate this, we studied blood samples from two large groups of pregnant women in Denmark and the United States. We measured hundreds of small molecules, known as metabolites, and looked at how they relate to body weight and pregnancy outcomes. We found that certain metabolites help explain the link between higher body weight and complications such as diabetes and preeclampsia. These results suggest that blood metabolite testing could one day improve pregnancy care by identifying women at higher risk early on.

Overweight and obesity are increasing in prevalence, with over half of the world population projected to live with overweight, or obesity, by 2035[1]. Accordingly, there is a rising incidence of maternal obesity, which has been associated with pregnancy complications such as gestational diabetes, preeclampsia and cesarean section[2,3]. Body mass index (BMI) is a widely used anthropometric measure of adiposity, but it does not capture the underlying metabolic heterogeneity associated with obesity. Recent studies have shown that individuals with a normal BMI can exhibit an obesity-

[1]COPSAC, Copenhagen Prospective Studies on Asthma in Childhood, Herlev and Gentofte Hospital, University of Copenhagen, Copenhagen, Denmark. [2]Channing Division of Network Medicine, Brigham and Women's Hospital, Harvard Medical School, Boston, MA, USA. [3]Section of Food Microbiology, Gut Health and Fermentation, Department of Food Science, University of Copenhagen, Copenhagen, Denmark. ✉e-mail: david.horner@dbac.dk

related metabolome and elevated risk of cardiometabolic disease, suggesting that BMI alone may underestimate metabolic risk[4]. These perturbations, reflected in BMI-associated metabolites, have thus been linked to adverse pregnancy[5] and child birth outcomes[6]. Metabolic perturbations can be reliably and quantitatively captured via the blood metabolome, which therefore has the potential to identify pregnancies at risk by detecting specific disruptions in metabolic homeostasis, providing a detailed assessment beyond anthropometric risk.

Whilst recent studies have shown that metabolic perturbations associated with high pre-pregnancy BMI improve prediction of pregnancy complications[7,8], there is a lack of understanding into which metabolites are mediating these associations, and such knowledge may be important for guiding targeted interventions. Moreover, several studies have identified metabolomic alterations associated with pregnancy complications such as pre-eclampsia[9] and gestational diabetes[10], including longitudinal analyses showing that metabolic changes during pregnancy are blunted in women with obesity[5], which may indicate an altered baseline metabolic state. However, many of these studies are limited by small sample sizes and do not perform external validation of their findings. Finally, most studies are only conducted cross-sectionally and thus the inference of findings do not account for the known pregnancy changes in metabolomic profiles throughout the gestational period[11].

Building on this rationale, we investigated the relationship between maternal BMI, BMI-associated metabolomic perturbations, and pregnancy complications across two large, well-characterised pregnancy cohorts. We find that a maternal blood metabolite profile measured at mid-pregnancy robustly predicts pre-pregnancy BMI and captures additional metabolic variation linked to the risk of pregnancy complications beyond BMI itself. This metabolite score is significantly associated with multiple pregnancy complications, including gestational diabetes, cesarean section, and induction of birth, and is enriched for metabolites involved in sphingolipid and vitamin A metabolism. Replication in an independent cohort confirms that BMI-related metabolic perturbations, particularly in late pregnancy, are stronger predictors of gestational diabetes and preeclampsia than measured BMI alone. Together, these findings demonstrate that BMI-associated metabolomic signatures provide mechanistic and predictive insight into obesity-related pregnancy complications.

## METHODS

### Study design
The primary analysis was conducted within the Copenhagen Prospective Studies on Asthma in Childhood (COPSAC2010) mother-child cohort, supplemented by validation analyses in the VDAART mother-child cohort. The COPSAC2010 cohort includes 736 mothers recruited during pregnancy[12]. Ethnicity was self-reported by the mothers as Danish (European descent) or other. In validation analysis, we used the large independent US-based VDAART mother-child cohort ($n = 881$)[13], to validate our blood metabolome modelling[14] against pregnancy complication outcomes.

### Validation of pregnancy complications and maternal pre-pregnancy BMI
In COPSAC2010, pregnancy complications, including gestational diabetes mellitus, preeclampsia, cesarean section, birth induction and maternal antibiotics usage at birth, were documented during scheduled visits and validated against registry data. Discrepancies were resolved by reviewing the medical records of both the mother and child. Information on pre-pregnancy weight was obtained from pregnancy records, with BMI calculated using height measurements taken at the research unit.

In the VDAART validation cohort, pregnancy complications were documented beginning at enrolment and continued until delivery, as described previously[13]. Maternal pre-pregnancy BMI was determined from electronic medical record review[15]. At monthly visits, short questionnaires were administered to pregnant mothers to survey pregnancy complications through self-report; additionally, research staff conducted

electronic medical record reviews monthly to monitor for pregnancy complications. At delivery, research staff collected information on the mode of delivery, administration of antibiotics during delivery, and anthropometric measures.

### Blood metabolome
Metabolomics data from plasma samples were obtained from mothers at the mid-pregnancy stage (24 weeks of gestation) within the COPSAC2010 cohort and at two time points (10–18 weeks and 32–38 weeks of gestation) in the VDAART cohort. During a visit to the research clinic, a blood sample was drawn into an EDTA tube, then centrifuged at $\sim 1800 \times g$ for 10 min to isolate the plasma. The resulting supernatant was preserved at $-80\,°C$ for subsequent analysis. Metabolon, Inc. (NC, USA) conducted the untargeted metabolomic analysis of these plasma samples for both the COPSAC2010 and VDAART cohorts.

For sample preparation, we utilised the MicroLab STAR® system from Hamilton Company for automation. Quality control was ensured by adding recovery standards to each sample before extracting metabolites with methanol. This extraction involved 2 min of vigorous shaking with a Glen Mills GenoGrinder 2000 and subsequent centrifugation to precipitate proteins. The resulting extract was divided into four aliquots for analysis on different LC-MS/MS platforms, dried using a TurboVap® (Zymark) to remove the organic solvent, and stored under nitrogen overnight before LC-MS/MS preparation.

The LC-MS/MS analysis employed an ACQUITY Ultra-Performance Liquid Chromatography (UPLC) system from Waters, Milford, USA, combined with a Q Exactive™ Hybrid Quadrupole-Orbitrap™ mass spectrometer featuring a heated electrospray ionisation (HESI-II) source, provided by ThermoFisher Scientific, Waltham, Massachusetts, USA. Sample extracts were prepared in specific solvent mixtures tailored to each of the four LC methods used: two reverse phase UPLC-ESI(+) MS/MS for hydrophilic and hydrophobic molecules, one reverse phase UPLC-ESI(−) MS/MS, and one HILIC UPLC-(−) MS/MS. Mass spectrometry alternated between full scan MS and data-dependent MSn scans with dynamic exclusion, covering a scan range from 70 to 1000 m/z for both ion modes.

For data collection and quality control, raw data underwent extraction and peak identification, followed by QC procedures. Semi-quantification of samples was based on the area under the curve method, with further details provided in our previously published work[16].

Data preprocessing involved excluding metabolites with more than 33% missing values. For the remaining missing data, random forest imputation was applied using the missForest R package (v1.5)[17]. This method does not assume data are missing at random and has been shown to outperform minimum value imputation in preserving the structure and variability of metabolomics data[18]. Before analysis, the metabolome data were log-transformed, centred, and scaled. The dataset for mothers at mid-pregnancy (24 weeks of gestation) included a total of 760 annotated metabolites for analysis.

We used two VDAART blood metabolome pregnancy time points (10–18 weeks of gestation and 32–38 weeks of gestation), measured on the same platform as the COPSAC2010 samples, to replicate our findings for pregnancy complications. At the 24-week pregnancy time point in COPSAC2010, we compared overlapping metabolites with those from early pregnancy (10–18 weeks of gestation) and late pregnancy (32–38 weeks of gestation) in VDAART, identifying 640 and 689 overlapping metabolites, respectively. Using metabolite models trained on COPSAC2010 towards maternal pre-pregnancy BMI, we predicted BMI-metabolite scores in VDAART after applying the same preprocessing steps (log-transformation, centring, and scaling). These scores were then used to assess associations with pregnancy complications in VDAART and to explore potential differences in predictive strength between early and late gestation. This approach allowed for temporal comparison while maintaining methodological independence between training and validation datasets.

## Information on covariates

We included covariates based on a directed acyclic graph constructed to guide covariate selection using causal reasoning[19]. In our multivariable analysis, we included the following covariates for COPSAC2010: social circumstances (the first principal component of household income, maternal education level, and maternal age), smoking during pregnancy, child sex and nutrient-derived pregnancy dietary patterns (principal components 1, 2, and 3), information pertaining to these pregnancy dietary patterns can be found in existing work[20,21]. Missing covariate data were imputed, based on the available covariate information available using the imputePCA function with one component from the missMDA (v1.18) R package.

Covariates used in multivariable analysis for VDAART included social circumstances (the first principal component of household income, maternal education level, and maternal age at birth), child sex, smoking during pregnancy and self-reported ethnicity. Missing covariate data were imputed, based on the available covariate informatio,n using the imputePCA function with one component from the missMDA (v1.18) R package.

## Statistical analysis

Multivariable logistic regression models were used to determine the associations of maternal pre-pregnancy BMI and associated metabolite scores on pregnancy complications. To provide temporal resolution, we applied the COPSAC2010-derived BMI-metabolite scores to VDAART samples at both early (10–18 weeks) and late (32–38 weeks) gestation, allowing us to compare associations across gestational stages and assess whether predictive strength varied by timing. All multivariable models were adjusted for the previously mentioned covariates. Maternal pre-pregnancy BMI and associated metabolite scores were scaled to enhance interpretation, so odds ratios and estimates are interpreted as per SD change. As the data is structured at the child level, we excluded one child from each twin pair to avoid duplicate pregnancy records and ensure independent observations.

We used the caret R package (v6.0.90) to establish maternal pre-pregnancy BMI metabolite scores via sparse partial least squares regression of the COPSAC2010 metabolomics datasets, which overlapped with respective VDAART timepoints, with maternal pre-pregnancy BMI as the response. To improve interpretability and reduce the risk of overfitting, we used single-component models, applying repeated (10 times) cross-validated (with 5 segments) predictions. After reviewing models that varied in sparsity (incremented by 0.1 from 0 to 1), we opted for the model with the lowest root-mean-standard error for cross-validation (RMSECV) as the selection criteria. We performed pathway enrichment analysis to identify overrepresented metabolic pathways among the 46 BMI-associated metabolites selected by sparse partial least squares regression. Annotation was based on the Metabolon-provided biochemical classifications, linking each metabolite to its respective sub- and super-pathways. Enrichment was assessed using hypergeometric testing, comparing the observed count of selected metabolites per sub-pathway to the expected count based on all measured metabolites. Fold enrichment, and p-values were computed, with false discovery rate and Bonferroni correction applied to account for multiple testing.

To elucidate the role of metabolites as potential biomarkers or mediators in the association between maternal pre-pregnancy BMI and pregnancy complications, we employed a systematic backward elimination strategy[20]. We prioritised outcomes for mediation analysis where both maternal pre-pregnancy BMI and BMI metabolite scores exhibited significant associations in both cohorts. This approach ensures that the mediation analysis focuses on outcomes where metabolomic factors may play a direct role in mediating the relationship between maternal BMI and pregnancy complications, thus providing insights into potential biological mechanisms. Our analysis approach was designed to iteratively exclude metabolites, aiming to identify those with the most profound mediating influence on the outcome. We used two multivariable models: one linear regression model linking a composite metabolite score to maternal pre-pregnancy BMI and covariates, and another logistic regression model

assessing its mediating role between maternal pre-pregnancy BMI and pregnancy complication outcomes, using the same covariate structure. Backward elimination was applied to the first model by iteratively removing individual metabolites from the composite score and recalculating the mediation effect. At each step, the metabolite whose removal most increased the average causal mediation effect was dropped, and the process continued until no further gain was observed. This causal mediation analysis was conducted with the mediation package in R (v4.5.0), with 10,000 simulations performed at each iteration. To formally compare model performance between the full maternal BMI-associated metabolite score and the subset of 16 mediating metabolites, we used a likelihood ratio test based on the chi-squared distribution to assess whether adding the subset score significantly improved model fit.

A significance level of 0.05 was used in all analyses, false-discovery rate control (FDR) applied where relevant (<0.05), and all statistical tests were two-sided. All data analyses were performed with the statistical software R version 4.1.1. Other R packages utilised in this analysis include tidyverse (v1.3.1) for general data processing and visualisation, dplyr (v1.0.10) for data wrangling, broom (v0.7.12) for tidying model outputs, lubridate (v1.8.0) for handling date variables, ggpubr (v0.4.0) for producing publication-ready plots, and tableone (v0.13.0) for generating descriptive baseline tables.

## Ethics statement

This study was carried out in compliance with the Declaration of Helsinki and received approval from the Danish Ethics Committee (H-B-2008-093) and the Danish Data Protection Agency (2015-41-3696). It was conducted and monitored according to Good Clinical Practice (GCP) requirements, as outlined in the EU Clinical Trials Directive (2001/20/EC) and the EU GCP Directive (2005/28/EC). Written informed consent was obtained from all participants before any study procedures. In the COPSAC2010 cohort, mothers provided written informed consent for both their own and their child's participation and for the use of de-identified data in future research. Participant confidentiality is safeguarded in line with GCP standards.Access to the Vitamin D Antenatal Asthma Reduction Trial (VDAART) mother–child cohort data was granted through a formal data-sharing agreement between the Copenhagen Prospective Studies on Asthma in Childhood (COPSAC) and VDAART investigators. The VDAART study was approved by the Institutional Review Boards of all participating clinical centres (Boston University Medical Centre, Washington University in St. Louis, and Kaiser Permanente Southern California). All participating mothers provided written informed consent for both their own and their children's data to be used for research. Analyses for the present study were conducted on de-identified data stored securely on the COPSAC server in accordance with these agreements and data-governance procedures. Because no new data were collected, and all analyses were performed on de-identified data consistent with the original VDAART ethical approvals and participant consents, additional IRB approval was not required. Data access is restricted to approved investigators and must occur in collaboration with VDAART investigators; requests can be directed to the VDAART leadership team (rejas@channing.harvard.edu).

## RESULTS

### Cohort characteristics

In the COPSAC2010 cohort, we included 690 mothers with data on maternal pre-pregnancy BMI. 684 mothers had untargeted metabolomics performed on blood samples taken at 24 weeks of gestation. Table 1 presents the baseline characteristics of the COPSAC2010 cohort, stratified by tertiles of maternal BMI. Significant differences were observed between the strata of maternal pre-pregnancy BMI, with higher BMI associated with lower maternal education levels, higher child birth weight, and a Western dietary pattern. Baseline characteristics differences were similar across strata of clinical obesity categories (Table S1). A descriptive summary of socio-demographic background and pregnancy complication outcomes across both mother-child cohorts can be found in Table S2.

**Table 1 | Model covariates stratified by low, middle, and high (tertiles) for maternal pre-pregnancy BMI in pregnant mothers in the COPSAC2010 cohort**

| Baseline characteristics | Lowest tertile | Middle tertile | Highest tertile | *p*-value |
|---|---|---|---|---|
| *n* = | 230 | 230 | 230 | |
| Male sex (%) | 113 (49.1) | 125 (54.3) | 115 (50.0) | 0.487 |
| White ethnicity (%) | 220 (95.7) | 222 (96.5) | 220 (95.7) | 0.862 |
| Income type (%) | | | | 0.466 |
| Low | 24 (17.8) | 16 (11.7) | 20 (12.4) | |
| Medium | 37 (27.4) | 40 (29.2) | 54 (33.5) | |
| High | 74 (54.8) | 81 (59.1) | 87 (54.0) | |
| Maternal education level at birth (%) | | | | <0.001 |
| Low | 11 (4.8) | 11 (4.8) | 28 (12.2) | |
| Medium | 134 (58.3) | 145 (63.0) | 159 (69.1) | |
| High | 85 (37.0) | 74 (32.2) | 43 (18.7) | |
| Maternal age at birth (mean (SD)) | 32.33 (4.32) | 32.37 (4.18) | 32.09 (4.55) | 0.754 |
| Birthweight (mean (SD)) | 3.47 (0.55) | 3.52 (0.55) | 3.65 (0.52) | 0.001 |
| Gestational age (mean (SD)) | 279.27 (10.98) | 278.35 (12.40) | 280.07 (10.74) | 0.273 |
| Cesarean section (%) | 41 (17.8) | 47 (20.4) | 60 (26.1) | 0.088 |
| Maternal smoking during pregnancy (%) | 15 (6.5) | 18 (7.8) | 21 (9.1) | 0.581 |
| Siblings (mean (SD)) | 1.52 (0.73) | 1.53 (0.82) | 1.62 (1.00) | 0.439 |
| Pregnancy varied dietary pattern (mean (SD)) | 0.14 (1.04) | −0.01 (0.92) | −0.08 (1.03) | 0.083 |
| Pregnancy Western dietary pattern (mean (SD)) | −0.15 (0.98) | −0.01 (1.04) | 0.12 (0.95) | 0.023 |
| Pregnancy PC3 dietary pattern (mean (SD)) | 0.15 (1.04) | 0.02 (0.93) | −0.15 (1.00) | 0.009 |
| Maternal pre-pregnancy BMI (mean (SD)) | 20.61 (1.18) | 23.64 (0.88) | 29.40 (4.00) | <0.001 |
| Maternal BMI metabolite score (mean (SD)) | −0.67 (0.75) | −0.19 (0.76) | 0.76 (0.90) | <0.001 |

Income type is categorised by household as low (<50,000 euros), medium (50,000–110,000 euros), and high (>110,000 euros). Maternal education level at birth is categorised as low (primary, secondary, or college graduate), medium (tradesman or bachelor's degree), and high (master's degree). Principal component 1 (PC1) represents a varied dietary pattern, principal component 2 (PC2) a Western dietary pattern, and principal component 3 (PC3) an additional dietary pattern. BMI denotes body mass index. Group comparisons use two-sided Pearson's χ² tests for categorical variables and two-sided one-way ANOVA for continuous variables.

## The maternal BMI blood metabolome

We utilised a supervised machine learning methodology, sparse partial least square modelling, on untargeted blood metabolomics sampling taken at 24 weeks of gestation in the COPSAC2010 cohort to find the metabolites that, in combination, best describe the response variable: pre-pregnancy BMI. 46 metabolites out of 640 potential metabolites available for modelling (7.2%) were selected in the model (RMSECV = 0.78, R2CV = 0.39). Loadings for these metabolites can be seen in Fig. 1 and Table S3. Further, to evaluate whether the relationship between predicted and measured BMI varied by pregnancy complication status, we visualised this association across maternal complication groups. Figure S1 shows the relationship between maternal BMI metabolite score and measured pre-pregnancy BMI, with women experiencing specific complications highlighted against the broader cohort. We performed pathway enrichment analysis on these 46 metabolites, considering 25 subpathways. This analysis revealed that

sphingomyelins and metabolites associated with Vitamin A metabolism were significantly enriched with higher pre-pregnancy BMI (FDR < 0.05) (Fig. S2). An analysis of missingness among selected metabolites was conducted to investigate whether the pattern of missing data was associated with specific pregnancy complications. Notably, only for individuals who underwent cesarean section, there was a lower likelihood of missing metabolite data (OR = 0.73, 95% CI [0.52–0.97], *p* = 0.040), specifically for the selected metabolites. This suggests that the missingness is not entirely random and highlights the importance of addressing this non-random missingness in our imputation approach, as shown in Table S4.

## Maternal BMI and the associated blood metabolite score are associated with pregnancy complications in COPSAC2010

Maternal pre-pregnancy BMI was significantly associated with several pregnancy complications in multivariable models adjusted for social circumstances, child sex, maternal smoking, and maternal dietary patterns (Table 2). Per standard deviation (SD) increase in BMI, the odds of gestational diabetes increased (OR 1.90 [1.29–2.74], $p = 6.75 \times 10^{-4}$), as did the odds of caesarean section (OR 1.23 [1.03–1.47], *p* = 0.023), induction of birth (OR 1.42 [1.21–1.67], $p = 2.86 \times 10^{-5}$), and maternal antibiotics at birth (OR 1.18 [1.01–1.39], *p* = 0.042). Corresponding estimates for per unit change in BMI are provided in Table S5. When considering acute or elective reasons for cesarean section, only emergency cesarean section was significantly associated with maternal BMI (OR 1.30 [1.05–1.61], *p* = 0.015) in comparison with elective cesarean section (OR 1.07 [0.81–1.37], *p* = 0.628).

The maternal BMI metabolite score, derived using sparse partial least squares modelling as described above (Fig. 1, Table S3), was significantly associated with several pregnancy complications. Per SD increase in the score, the odds of gestational diabetes increased (OR 2.47 [1.45–4.24], $p = 8.51 \times 10^{-4}$), as did the odds of caesarean section (OR 1.29 [1.06–1.58], *p* = 0.011), induction of birth (OR 1.31 [1.10–1.56], *p* = 0.002), and maternal antibiotics at birth (OR 1.21 [1.01–1.44], *p* = 0.035), with slightly higher estimates than for maternal BMI alone (Table 2). Additionally, the maternal BMI metabolite score was positively and significantly associated with pre-eclampsia (OR 1.54 [1.04–2.26], *p* = 0.030), whereas maternal BMI alone was not (OR 1.22 [0.86–1.67], *p* = 0.237). Again, only emergency cesarean section was associated with the maternal BMI metabolite score (OR 1.38 [1.07–1.77], *p* = 0.012), compared to elective cesarean section (OR 1.12 [0.85–1.47], *p* = 0.420).

## COPSAC2010 and VDAART cohort comparison and baseline characteristics

We sought replication of our findings in VDAART, a large US-based mother-child cohort. We included participants with corresponding pregnancy blood metabolome and pre-pregnancy BMI data (*n* = 775). There were significant differences in baseline characteristics between the VDAART and COPSAC2010 cohorts. The VDAART cohort had lower income, lower maternal education levels, younger maternal age, lower gestational age at birth, lower rates of maternal smoking, a higher proportion of non-white participants, and lower birth weight (*p* < 0.001). Additionally, the VDAART cohort had a significantly higher mean pre-pregnancy BMI (28.4 vs. 24.6, *p* < 0.001) and differences in incidence of pregnancy complications, including gestational diabetes (5.2% vs. 2.2%), cesarean section delivery (29.5% vs. 21.6%), induction of birth (18.2% vs. 35.6%), and administration of maternal antibiotics at birth (38.7% vs. 32.2%) (Table S2). Supplementary Fig. S3 shows the metabolomic variability in COPSAC2010 and VDAART, illustrating closely matched variances across pregnancy samples.

## Validation of metabolite scores and their association with pregnancy complications

Maternal pre-pregnancy BMI was significantly associated with several pregnancy complications in the VDAART cohort in multivariable models adjusted for social circumstances, child sex, and maternal smoking (Table 2). Per standard deviation (SD) increase in BMI, the odds of

**Fig. 1 | This figure illustrates the results from a supervised machine learning analysis (sparse partial least squares modelling) applied to untargeted blood metabolomics data collected at 24 weeks gestation in the COPSAC2010 cohort (*n* = 684).** Out of 640 potential metabolites, 46 were selected for their ability to predict maternal pre-pregnancy BMI. The bar plot ranks these metabolites by their scores, with colours representing different metabolite pathways and patterns indicating whether they mediate the association between maternal BMI and gestational diabetes (striped for mediators, solid for non-mediators). Pathway enrichment analysis revealed significant enrichment in sphingomyelin and Vitamin A metabolism (FDR < 0.05). Some metabolites, such as carotene diol and gentisate, appear protective, while others, such as ceramide and sphingomyelin, may increase gestational diabetes risk.

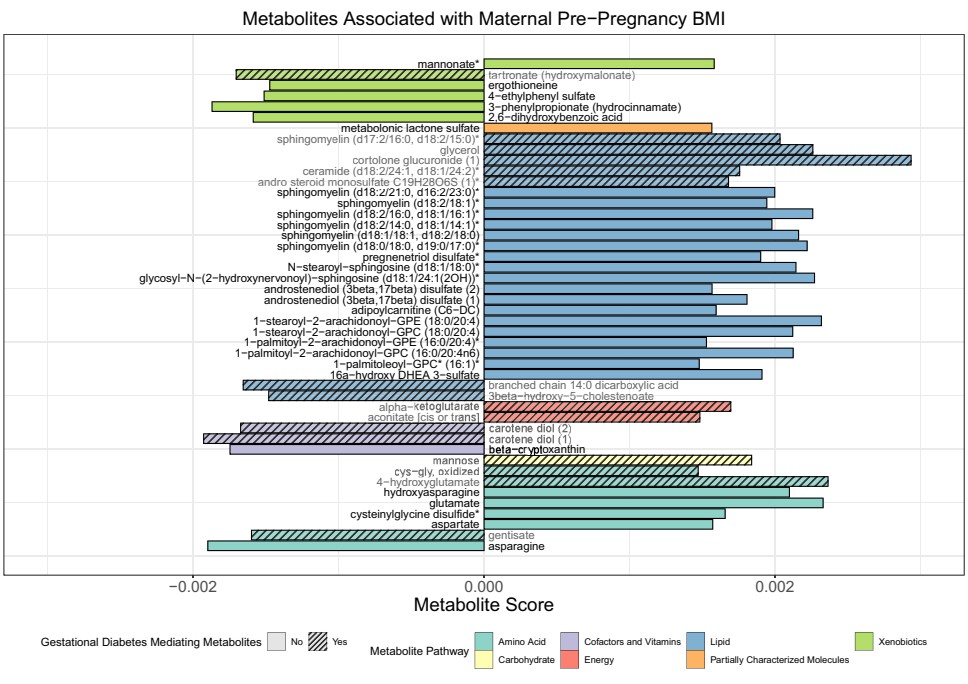

## Table 2 | Multivariable logistic regression associations of maternal pre-pregnancy BMI and BMI-linked metabolite scores with pregnancy complications

| COPSAC2010 pregnancy complications | Maternal BMI | Maternal BMI 24 week metabolite score | - |
|---|---|---|---|
| Gestational diabetes | 1.90 [1.29–2.74] ($p = 6.75 \times 10^{-4}$) | 2.47 [1.45–4.24] ($p = 8.51 \times 10^{-4}$) | – |
| Preeclampsia | 1.22 [0.86–1.67] ($p = 0.237$) | 1.54 [1.04–2.26] ($p = 0.030$) | – |
| Caesarean section | 1.23 [1.03–1.47] ($p = 0.023$) | 1.29 [1.06–1.58] ($p = 0.011$) | – |
| Emergency | 1.30 [1.05–1.61] ($p = 0.015$) | 1.38 [1.07–1.77] ($p = 0.012$) | – |
| Elective | 1.07 [0.81–1.37] ($p = 0.628$) | 1.12 [0.85–1.47] ($p = 0.420$) | – |
| Induction of birth | 1.42 [1.21–1.67] ($p = 2.86 \times 10^{-5}$) | 1.31 [1.1–1.56] ($p = 0.002$) | – |
| Maternal antibiotics at birth | 1.18 [1.01–1.39] ($p = 0.042$) | 1.21 [1.01–1.44] ($p = 0.035$) | – |
| VDAART pregnancy complications | Maternal BMI | Maternal BMI 10–18 week metabolite score | Maternal BMI 32–38 week metabolite score |
| Gestational diabetes | 1.49 [1.08–2.02] ($p = 0.012$) | 1.51 [1.08–2.14] ($p = 0.018$) | 2.1 [1.48–3.03] ($p = 4.97 \times 10^{-5}$) |
| Preeclampsia | 1.11 [0.76–1.55] ($p = 0.553$) | 1.4 [0.99–2] ($p = 0.060$) | 1.82 [1.23–2.72] ($p = 0.003$) |
| Caesarean section | 1.36 [1.14–1.61] ($p = 5.60 \times 10^{-4}$) | 1.17 [1–1.38] ($p = 0.058$) | 1.15 [0.98–1.35] ($p = 0.095$) |
| Emergency | 1.36 [1.06–1.72] ($p = 0.014$) | 1.25 [0.97–1.62] ($p = 0.081$) | 1.11 [0.86–1.44] ($p = 0.410$) |
| Elective | 1.23 [1.02–1.49] ($p = 0.033$) | 1.09 [0.91–1.31] ($p = 0.367$) | 1.13 [0.94–1.36] ($p = 0.192$) |
| Induction of birth | 1.30 [1.07–1.57] ($p = 0.008$) | 1.2 [0.99–1.46] ($p = 0.062$) | 1.21 [1–1.47] ($p = 0.048$) |
| Maternal antibiotics at birth | 1.11 [0.94V1.31] ($p = 0.196$) | 1.09 [0.94–1.27] ($p = 0.267$) | 1.01 [0.87–1.17] ($p = 0.917$) |

COPSAC2010 metabolite scores are from 24-week samples; VDAART scores are from 10–18- and 32–38-week samples. Entries are Odds ratios (95% confidence limits, *p*-value) per 1 SD increase in the predictor (BMI and metabolite scores standardised within cohort/visit), making estimates comparable. *P*-values are from two-sided Wald tests. No adjustments were made for multiple comparisons.

gestational diabetes increased (OR 1.49 [1.08–2.02], *p* = 0.012), as did the odds of caesarean section (OR 1.36 [1.14–1.61], *p* = 5.60 × 10⁻⁴) and induction of birth (OR 1.30 [1.07–1.57], *p* = 0.008), whereas no significant associations were observed for preeclampsia (OR 1.11 [0.76–1.55], *p* = 0.553) or maternal antibiotics at birth (OR 1.11 [0.94–1.31], *p* = 0.196).

To test the generalisability of our metabolite-based BMI models, we applied the 46-metabolite COPSAC2010-trained prediction scores to metabolomics data obtained from maternal blood samples in VDAART at the gestational windows: early (10–18 weeks) and late pregnancy (32–38 weeks). These scores were strongly associated with actual maternal pre-pregnancy BMI (10–18 weeks: β 4.62, *p* = 10⁻⁶³; 32–38 weeks: β 4.26, *p* = 10⁻⁵⁴), validating the metabolite signature's consistency across gestation

and independent cohorts. Figure S4 illustrates the relationship between the maternal BMI metabolite score and measured pre-pregnancy BMI in the VDAART cohort in early gestation (10–18 weeks), with women experiencing specific pregnancy complications highlighted against the broader cohort.

Maternal BMI metabolite scores in VDAART provided nuanced insights into pregnancy complications at different gestational stages. The associations between each of the 46 BMI-related metabolites at both pregnancy timepoints and the risk of gestational diabetes and preeclampsia were examined and can be found in Table S6. Early-pregnancy BMI metabolite scores (10–18 weeks) demonstrated weaker associations with gestational diabetes (OR 1.51 [1.08–2.14], *p* = 0.018) compared to late-pregnancy

scores (32–38 weeks) (OR 2.10 [1.48–3.03], $p < 0.001$). Similarly, late-pregnancy metabolite scores were significantly associated with preeclampsia (OR 1.82 [1.23–2.72], $p = 0.003$), while early-pregnancy scores were not significantly associated with preeclampsia (OR 1.40 [0.99–2.00], $p = 0.060$).

Associations between metabolite scores and delivery-related outcomes were less consistent. Metabolite scores were not significantly associated with cesarean section, though trends were observed for both early and late pregnancy. Induction of birth was associated with the late-pregnancy score ($p = 0.048$) and showed a borderline association with the early-pregnancy score ($p = 0.062$). No associations were found for maternal antibiotics at birth (Table 2). The lack of consistent associations with cesarean section, induction of birth, or maternal antibiotics in our external validation may indicate that the inferences observed in COPSAC2010 might be more related to the direct mechanical consequences of high BMI rather than the metabolic perturbations disturbances associated with BMI.

### Independence of the maternal BMI associated metabolite scores associations with pregnancy complications

Next, we assessed whether the metabolite scores had independent predictive value over anthropometrically measured BMI in VDAART (Table S7). When adjusting the maternal metabolite BMI scores for maternal BMI, the late pregnancy metabolite score demonstrated some clear distinctions in predictive capacity compared to early pregnancy. The association between the BMI-adjusted metabolite score at 32–38 weeks and gestational diabetes (OR 1.91 [1.24–2.99], $p = 0.004$) underscores the significance of metabolic changes later in pregnancy, as the early-pregnancy metabolite score was not significantly associated with gestational diabetes (OR 1.27 [0.82–1.96], $p = 0.285$). Similarly, for preeclampsia, the 32–38 week BMI-adjusted metabolite score showed a significant association (OR 2.12 [1.32–3.47], $p = 0.002$), indicating a stronger predictive value compared to the early-pregnancy score (OR 1.49 [0.93–2.39], $p = 0.098$). These results suggest that late-pregnancy metabolic perturbations may be more informative for predicting preeclampsia.

### Mediating role and independent predictive value of metabolite scores in pregnancy complications

To explore whether any of the 46 BMI-associated metabolites causally mediate the association between maternal BMI and gestational diabetes, an association we found in both cohorts, we conducted a backward elimination mediation analysis. This analysis focused on identifying a subset of mediating metabolites within the COPSAC2010 cohort and thereafter assessing inference of these metabolites in VDAART. Our multivariable mediation analysis identified 16 (35%) metabolites (Fig. 1, Table S3) that mediated the association between maternal BMI and gestational diabetes. Among these 16 metabolites, a clear pattern emerged related to dietary and lipid metabolism. Metabolites such as carotene diol and gentisate, derived from plant sources, may confer some protection from gestational diabetes risk, evidenced by their negative associations. In contrast, lipid-related metabolites such as ceramide (d18:2/24:1, d18:1/24:2) and sphingomyelin (d17:2/16:0, d18:2/15:0) exhibit positive associations, suggesting their potential role in exacerbating gestational diabetes risk.

In the validation cohort, we predicted a metabolite score using the subset of mediating metabolites identified in COPSAC2010. This score was incorporated alongside the full-metabolite score of 46 BMI-associated metabolites in our modelling. Whilst the full maternal BMI score was significant in early gestation (OR 1.51 [1.08–2.14], $p = 0.018$), the subset of mediating metabolites showed an even stronger association with gestational diabetes here (OR 1.81 [1.29–2.56], $p = 6.62 \times 10^{-4}$). The mutually adjusted model indicated a significant improvement in predictive capacity with a $p$-value of 0.009. Similarly, whilst findings were significant with the full maternal BMI score in late gestation (OR 2.10 [1.48–3.03], $p = 4.97 \times 10^{-5}$), the subset of mediating metabolites had a stronger association (OR 2.26 [1.61–3.22], $p = 3.67 \times 10^{-6}$). The mutually adjusted model also demonstrated a significant improvement in predictive capacity, with a $p$-value of 0.016. These findings suggest that the subset of mediating metabolites not

only retained but enhanced the predictive power for gestational diabetes compared to the full-metabolite score alone, particularly when modelled alongside maternal BMI.

## DISCUSSION

Our study investigated the complex relationship between maternal BMI and pregnancy complications through blood metabolomics in two large, well-characterised mother-child cohorts, COPSAC2010 and VDAART. By employing untargeted blood metabolomics and supervised machine learning techniques, we identified specific metabolic perturbations linked to higher maternal pre-pregnancy BMI that may contribute to adverse pregnancy outcomes. In external validation, our findings suggest that BMI-associated metabolites may enhance the risk assessment of pregnancy complications, for complications such as gestational diabetes and pre-eclampsia, beyond traditional anthropometric measures. Furthermore, our results indicate stronger predictive power in late gestation compared to early gestation, highlighting the importance of gestational timing in metabolic risk assessments. Finally, our mediation analysis pinpointed specific metabolites that mediate the association between maternal BMI and gestational diabetes, providing mechanistic insights. Notably, even when accounting for the full BMI-associated metabolite score in modelling, considering a subset of these potentially causal metabolites allowed for greater inference, providing further evidence of potential causality. These insights may help guide the development of targeted therapeutic interventions and highlight the feasibility of integrating metabolomic screening into routine prenatal care for personalised risk assessment and management.

Many of our findings align with existing literature, which suggests that metabolomic profiles in pregnancy may enhance predictive capacity for pregnancy outcomes[7,8,22], particularly in late pregnancy compared to early pregnancy. This complements our observation in the VDAART cohort, where BMI-metabolite scores showed stronger inference in later pregnancy than in early pregnancy. Supporting this, a study predicting healthy pregnancies (absence of pregnancy complications) found that early-pregnancy metabolites did not significantly improve predictive models beyond maternal characteristics alone[23]. Late gestation has been suggested to be a period of metabolic stress, exacerbated by the physiological insulin-resistant state[24], but of course also closer to any adverse pregnancy outcomes[25].

Several previous studies have used multivariate approaches to define metabolic components associated with maternal BMI, including early pregnancy profiles[6,25,26]. Although our study employed an untargeted platform and derived BMI-metabolite scores using external validation, we similarly observed enrichment of lipid-related metabolites, including ceramides and fatty acids, as well as specific amino acid and vitamin metabolism pathways. This overlap supports the reproducibility of BMI-associated metabolic perturbations in pregnancy and highlights their potential generalisability across pregnancy populations and analytical platforms. In our multivariable mediation analysis, we identified 16 metabolites mediating the association between maternal BMI and gestational diabetes. A distinct pattern associated with dietary and lipid metabolism emerged among these 16 metabolites. In line with our findings, ceramide species have been identified in several studies as potential biomarkers for gestational diabetes mellitus. For example, C18:1-Ceramide has been proposed as a potential biomarker for early gestational diabetes detection during the first trimester, demonstrating significant differences between pregnant women with gestational diabetes and those with normal glucose tolerance[27]. Sphingolipids, especially ceramides, may contribute to diabetes[28]. Accordingly, sphingomyelin profiles in placentas show significant differences between pregnancies with and without gestational diabetes[29]. Carotene diols from Vitamin A metabolism have been shown to be negatively associated with gestational diabetes, suggesting their potential beneficial effect in reducing gestational diabetes risk. Carotene diol metabolic pathways have been shown as potential mediators, negatively associated with maternal obesity[30], consistent with our findings.

The reliability of our findings made in the COPSAC2010 cohort was confirmed through external validation in the VDAART cohort, wherein the

maternal BMI metabolite score and anthropometric BMI were strongly correlated, and our metabolome modelling provided additional benefit in predicting pregnancy complications. Our work aligns with previous work[9], where the addition of metabolites, measured using the same platform, improved prediction of pregnancy-related disorders, including gestational diabetes and preeclampsia, beyond established risk factors. Notably, glycerol and 4-hydroxyglutamate were among the top predictors for these outcomes and were also retained in our maternal BMI model. In our analysis, both of these metabolites significantly mediated the association between maternal BMI and gestational diabetes. Similarities in metabolic profiles between these conditions may be explained by underlying pathophysiologic processes such as insulin resistance, low-grade inflammation, oxidative stress, and endothelial dysfunction[31], along with the coexistence of obesity and pregnancy disorders. Previous research has shown that pregnancy-related metabolic changes are smaller in women with obesity, who display metabolic perturbations already in early pregnancy[5]. This suggests that the metabolic alterations of obesity and pregnancy disorders share a common origin and supports our observation that higher baseline BMI levels in VDAART enhance the predictive power of metabolite scores, even in early gestation. This interpretation is in line with the concept of pregnancy as a physiological stress test, which may unmask underlying metabolic vulnerability in women predisposed to cardiometabolic disease later in life[32]. Moreover, research has shown the potential for maternal BMI to moderate the relationship between specific risk-metabolites and pregnancy complications such as preeclampsia and underscores the necessity to account for different maternal risk profiles and associated biomarker profiles[33]. Taken together, these findings support the clinical relevance of our approach. While the observational nature of our study limits definitive conclusions about temporality, maternal BMI and metabolomic profiling generally preceded the onset of complications, supporting their potential utility for early risk stratification.

Our study has several strengths. First, we used untargeted blood metabolomics and supervised machine learning techniques to identify specific metabolic perturbations linked to higher maternal BMI, which allowed for a detailed understanding of the metabolic contributions to adverse pregnancy outcomes. Second, we validated our findings in an independent cohort, VDAART, which differed in many characteristics from our initial COPSAC2010 cohort, confirming the robustness and generalisability of our results. Third, our study design included a large, well-characterised sample with a comprehensive set of metabolites measured across all gestational periods, allowing for a comparative assessment and enhancing the reliability of our findings. However, our study also has limitations. While we observed improved predictive capacity with BMI-associated metabolites, we did not benchmark our risk prediction models against established clinical screening tools currently used in practice for gestational diabetes or preeclampsia. Additionally, differences in baseline characteristics between cohorts, such as maternal education and income, highlight the need for careful interpretation of results across diverse populations. Finally, the study was conducted in high-resource settings, which may limit the generalisability of our findings to lower-resource environments.

In conclusion, our study highlights the complex interplay between maternal BMI and pregnancy complications through detailed metabolomic analysis in two diverse cohorts. By identifying specific metabolic perturbations associated with higher maternal BMI, we have demonstrated the potential of metabolomic profiling to enhance the prediction of adverse pregnancy outcomes beyond traditional measures. Additionally, our mediation analysis provides mechanistic insights that could guide targeted therapeutic interventions. These findings underscore the potential for integrating metabolomic screening into prenatal care to improve personalised risk assessment and clinical care. Future research should validate these results in diverse populations and compare them directly with current clinical practices to ensure broader applicability.

## Data availability

Participant-level personally identifiable data are protected under the Danish Data Protection Act and European Regulation 2016/679 of the European Parliament and of the Council (GDPR) that prohibit distribution even in pseudo-anonymized form. Data from the Vitamin D Antenatal Asthma Reduction Trial (VDAART) are additionally protected under U.S. data privacy regulations and participant consent agreements. However, participant-level data from both cohorts can be made available under a formal data-transfer agreement as part of a collaborative effort with COPSAC and VDAART investigators. Requests for data access should be directed to the COPSAC data manager (ulrik.ralfkiaer@dbac.dk) and the VDAART leadership team (rejas@channing.harvard.edu). The source data for Fig. 1 are provided in the Supplementary Data.

## Code availability

The custom code employed in this research is freely accessible to the public for transparency and reproducibility purposes. Code for the metabolite-signature derivation and the backward-elimination mediation framework is available via the following link: https://github.com/dlghorner/Code-for-Maternal-BMI-Article-Communications-Medicine.

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

## Acknowledgements

We express our deepest gratitude to the families of the COPSAC2010 and VDAART cohorts for all their support and commitment. We acknowledge and appreciate the unique efforts of the COPSAC research team. We acknowledge all funding received by COPSAC, listed on www.copsac.com. The Lundbeck Foundation (Grant no R16-A1694 and R269-2017-5), The Ministry of Health (Grant no 903516), Danish Council for Strategic Research (Grant no 0603-00280B) and The Capital Region Research Foundation have provided core support to the COPSAC research centre. This project has received funding from the European Research Council (ERC) under the European Union's Horizon 2020 research and innovation programme (grant agreement No. 946228) (BC). MAR is funded by the Novo Nordisk Foundation (Grant no NNF21OC0068517). JL-S (R01HL123915, R01HL155742, and R01HL141826) and SHC (K01HL153941) are funded through the National Institute of Heart, Lung, and Blood Institute.

## Author contributions

D.H. has written the first draft of the manuscript. All co-authors (R.V., T.W., M.A., M.L., N.P., J.L.S., K.B., J.S., and B.C.M.R.) have provided important intellectual input and contributed considerably to the analyses and interpretation of the data. All authors guarantee that the accuracy and integrity of any part of the work have been appropriately investigated and resolved, and all have approved the final version of the manuscript. The last author, M.R., had full access to the data. The corresponding author had final responsibility for the decision to submit for publication.

## Competing interests

J.L.-S. is a scientific advisor for Precion Inc and a consultant to Tru Diagnostic, Inc. Remaining authors (D.H., R.V., T.W., M.A., M.L., N.P., J.L.S., K.B., J.S., B.C., and M.R.) declare no potential, perceived, or real conflict of interest regarding the content of this manuscript. The funding agencies did not have any role in the design and conduct of the study; collection, management, and interpretation of the data; or preparation, review, or approval of the manuscript. No pharmaceutical company was involved in the study.
