## [Transparent Peer Review file · Communications Medicine]

A Metabolomic Signature of Maternal BMI is Associated with Pregnancy Complications Across Two Independent Pregnancy Cohorts

Corresponding Author: Dr David Horner

Version 0:

Reviewer comments:

Reviewer #1

(Remarks to the Author)

This study addresses important research questions and benefits from a rich dataset. However, it would be strengthened by situating itself more explicitly within the existing literature and clearly highlighting its novel contributions.

Findings of prior research on the associations between BMI and pregnancy complications should be discussed in the background. Several studies have identified metabolic profiles of BMI and pregnancy complications during pregnancy, including longitudinal associations. Kivelä et al. (cited in the manuscript) have focused on longitudinal associations and identified metabolic changes in pregnancy complications that were independent of BMI. The authors should build their study's justification taking into account this existing evidence, not just dismissing it as inadequate.

Another question is clinical relevance: while associations between BMI-related metabolic changes may predict pregnancy complications better when pregnancy complications have already occurred, from the clinical point of view prediction of pregnancy complications is important before their onset.

It is also counterintuitive that the discovery cohort includes only one measurement point, while longitudinal analysis is conducted in a replication cohort – the authors need to justify this approach.

Statistical methods require more detail. The authors should specify which research question was addressed with each method. Longitudinal analysis is not explained in the methods and it is not clear based on what the conclusions that the associations were stronger towards late gestation were made.

Finally, several studies (including those cited by the authors, as well this Pubmed 35948657) have used similar analytical approaches to identify metabolic profiles of BMI during pregnancy. However, the authors do not discuss how the metabolic profile of BMI identified in their sample align to these previous findings.

Reviewer #2

(Remarks to the Author)

Version 1:

Reviewer comments:

Reviewer #1

(Remarks to the Author)

Thank you for thoroughly addressing the comments. I recommend this manuscript for publication.

Reviewer #2

(Remarks to the Author)

REVISION SUMMARY (REVIEWER #2):

Horner et al. engaged constructively with the reviewers' comments and questions and made changes to their manuscript accordingly. The data presentation and general outlay of the manuscript markedly improved. The authors are commended for taking on the reviewers' critiques with rigor. Yet, some deficiencies remain: in the first review iteration, the reviewer did not spot the absence of correction for multiple testing when evaluating BMI and metabolite score associations with the multiple pregnancy outcomes, increasing the likelihood of false positive findings. No justification for not applying multiple testing correction was given. Some further recommendations / comments are therefore provided below.

RESEARCH SUMMARY:

In their research Horner et al used metabolomics data as derived from blood samples taken at 24 weeks of pregnancy from participants of the COPSAC2010 study, to firstly select metabolite to generate a metabolite score to predict pre-pregnancy BMI and then to compare this score with pre-pregnancy BMI in terms of its association with adverse outcomes. In a second step, the researchers applied mediation analysis to refine their metabolite panel and to select a subset of metabolites which potentially indirectly mediate the effect of BMI, an imperfect measure of obesity, on gestational diabetes (GDM), resulting in a second metabolite score.

Metabolomics data as derived from blood samples at 2 gestational age windows (10-18 wks; 32-38 wks) from a second, independent pregnancy cohort VDAART were used to validate the metabolic scores in a variety of model permutations. The metabolic scores outperformed pre-pregnancy maternal BMI in predicting GDM and pre-eclampsia, with indications that these scores can be used to improve the prediction of adverse pregnancy outcomes when combined with pre-pregnancy BMI. This suggest that the total effect of BMI on pregnancy outcome has a direct and an indirect effect component. The authors also find that the metabolic scores prediction ability increase with gestational age.

RESEARCH IMPACT:

The research of Horner et al. add to the concept that BMI is an imperfect measure of obesity, and therefore maternal pre-pregnancy BMI is a limited predictor of adverse pregnancy outcomes. With their research Horner et al show that specific panel of metabolites have the ability to complement pre-pregnancy BMI, possibly through a better capture of obesity, and thus improve prediction of pregnancy outcomes.

REVISION REVIEW

1. Abstract

a. Reviewer comments #6 - #10: Addressed appropriately

2. Introduction

a. Reviewer comments # 11 - #14: Addressed appropriately

3. Results

a. Reviewer comments #15 - #16: Addressed appropriately

b. Reviewer comment #17: Reviewer accepts the authors' modelling choice rationalisation; it is noted that the equivalent of Figure S4 for VDAART 32 – 38 wks is not generated. Given that prediction of the metabolite panel at this time point is most performant, the authors may want to add this Figure as well.

c. Reviewer comments #18 - #19: Addressed appropriately

d. Reviewer comments #18 - #19: Addressed appropriately

e. Reviewer comment #20: The authors are thanked for provision of the missingness list. The original question remains: is there a pattern in missingness for the 6 xenobiotic metabolites of the 46 panels across the outcomes.

f. Reviewer comments #21 - #27: Addressed appropriately

g. Reviewer comment #28: Given that there was no correction applied for multiple testing, the reviewer would still advise not to spend any discussion space on the association between the metabolite scores with the delivery-associated outcomes other than the conclusion. It is expected that upon correcting for multiple testing, no significant associations will remain in COPSAC2010 as well.

h. Reviewer comments #29-32: Addressed appropriately

i. Reviewer comment #33: Ok. Inference that the 16 metabolite model is more performant for predicting adverse outcomes than the 46 metabolite model (trained to predict BMI, not the adverse outcomes) may be done on the respective differences in reported odds ratios without the need for making the "detour" via the mutually adjusted model. But the reviewer is ok to follow the authors in this matter.

4. Discussion

a. Reviewer comment #34: The authors are thanked to provide the Vitamin D analysis.

b. Reviewer comment #35: Thanks for clarifying; this is line with Ottoson et al.

c. Reviewer comment #36: Addressed appropriately

5. Acknowledgments

a. Reviewer comment #37: Addressed appropriately

6. Methods

a. Reviewer comments #38-42: Addressed appropriately

REVIEW OF REVISED MANUSCRIPT:

1. Abstract

a. OK

2. Introduction

- a. Lines 59-60: Suggested rephrasing - "We also performed mediation analysis for these scenarios where both BMI and BMI-associated metabolite scores associate with adverse pregnancy outcomes to elicit specific metabolites which likely mediate the effect between maternal BMI and adverse pregnancy outcomes."
- b. Line 61: Replace "contrasting" by "evaluating"
- c. Line 63: extend the sentence as follows "...for predicting maternal health outcomes across gestation."

3. Results

- a. Line 78: the number of metabolites of 640 considered for analysis is based on their common presence in the COPSAC2010 and VDAART datasets, this is not apparent from the language in the results; it is only mentioned in the methods. Consider rephrasing as follows "...46 metabolites out of 640 metabolites available for modelling..."
- b. Lines 86-92; Consider removal of section discussing assumed non-randomness of missingness altogether (remove Table S4 as well; if retained, the heading of column 2 should be corrected (remove "p-value"). When correcting for multiple testing (at least 5 comparisons are made: GDM, PE, C-section, induction, antibiotics), the observed association will not reach significance. Here and in all other outcome analyses, the authors should correct for multiple testing.
- c. Lines 95 -96: Update results upon application of multiple testing correction; it is expected that several of the associations will be rendered non-significant, simplifying interpretation of the results.
- d. Lines 104 -113: Update results upon application of multiple testing correction; it is expected that several of the associations will be rendered non-significant, simplifying interpretation of the results.
- e. Line 119: correct ... "higher rates of maternal smoking..." to "lower rates of maternal smoking..."
- f. Lines 127 – 133: Update results upon application of multiple testing correction
- g. Line 135: consider rephrasing as follows: "...prediction scores to metabolomics data obtained from maternal blood samples..."
- h. Line 139: Clarify figure S4 is plotting the BMI-associated metabolite vs BMI for the 10-18 weeks gestation. Consider adding an additional figure plotting the data for the 32- 38 weeks gestation as well.
- i. Line 144: Replace "multivariable modelling" by "logistic regression analysis"
- j. Line 146-148: 1) Simplify the sentence as follows: "The associations between each of the 46...(Table S6) were also examined. 2) Move sentence to line 150 and put it prior to the sentence "Early pregnancy metabolite scores..." 3) add to caption of Table S6 that the single metabolite association analysis is exploratory and that no multiple testing correction was applied.
- k. Lines 149 – 154: Update results upon application of multiple testing correction; same conclusions expected
- l. Lines 165 – 175: Update results upon application of multiple testing correction; same conclusions expected
- m. Line 168 and Line 172: change "metabolite score" to "BMI-adjusted metabolite score"
- n. Line 174: replace "crucial" with "informative"
- o. Lines 184-185: rephrase as follows "..., derived from plant sources, may confer some protection from GDM risk, evidenced by..."
- p. Line 189: replace "analysis" with "cohort"
- q. Lines 193 – 198: addressed in above commentary to reply to reviewer comment #33. Additional question: does the 16 metabolite score also improves preeclampsia prediction?

4. Discussion

- a. Line 225: extend sentence as follows: "also closer to any adverse pregnancy outcomes occurring."
- b. Line 232: replace "cohorts" with "pregnancy populations"
- c. Line 234: Start sentence as follows: In line with our findings, ceramide species....
- d. Line 240: more accurate (?): ... show significant differences between pregnancies with and without gestational diabetes.
- e. Line 245: Consider rephrasing as follows: " The reliability of our findings made in the COPSAC2010 cohort was confirmed through external validation in the VDAART cohort wherein the maternal BMI ..."
- f. Lines 247 – 253: this section of the discussion appears in conflict with the results as presented in lines 104-113?
- g. Lines 282 – 284: the mentioned comparators are diagnostic criteria. More appropriate risk biomarker comparators are HbA1c for gestational diabetes screening in first trimester or PIGF and s-Flt1/PIGF for respectively early pregnancy and late pregnancy preeclampsia risk screening.

5. References

- a. Reference 19: no authors are listed
- b. Reference 20: no authors are listed

Version 2:

Reviewer comments:

Reviewer #2

(Remarks to the Author)

REVISION SUMMARY (REVIEWER #2):

Once more Horner et al. engaged constructively with the reviewer's comments and questions and made changes to their manuscript accordingly. Where Horner et al. did not implement changes as suggested by the reviewer, their rationale for not doing so is appropriately justified.

Overall, the revised manuscript now presents a compelling record of Horner et al. research with some relevant findings reported. Reviewer #2 has no further scientific comments / questions and wishes Horner et al. Some final minor editorial comments are provided below.

RESEARCH SUMMARY:

In their research Horner et al used metabolomics data as derived from blood samples taken at 24 weeks of pregnancy from participants of the COPSAC2010 study, to firstly select metabolite to generate a metabolite score to predict pre-pregnancy BMI and then to compare this score with pre-pregnancy BMI in terms of its association with adverse outcomes. In a second step, the researchers applied mediation analysis to refine their metabolite panel and to select a subset of metabolites which potentially indirectly mediate the effect of BMI, an imperfect measure of obesity, on gestational diabetes (GDM), resulting in a second metabolite score.

Metabolomics data as derived from blood samples at 2 gestational age windows (10-18 wks; 32-38 wks) from a second, independent pregnancy cohort VDAART were used to validate the metabolic scores in a variety of model permutations. The metabolic scores outperformed pre-pregnancy maternal BMI in predicting GDM and pre-eclampsia, with indications that these scores can be used to improve the prediction of adverse pregnancy outcomes when combined with pre-pregnancy BMI. This suggest that the total effect of BMI on pregnancy outcome has a direct and an indirect effect component. The authors also find that the metabolic scores prediction ability increase with gestational age.

RESEARCH IMPACT:

The research of Horner et al. adds to the concept that BMI is an imperfect measure of obesity, and therefore maternal pre-pregnancy BMI is a limited predictor of adverse pregnancy outcomes. With their research Horner et al show that specific panel of metabolites have the ability to complement pre-pregnancy BMI, possibly through a better capture of obesity, and thus improve prediction of pregnancy outcomes.

REVISION REVIEW

OK, Revision #2 comments are answered thoughtfully and with appropriate justification.

REVIEW OF REVISED MANUSCRIPT:

- Line 133 – replace: UPLC(-) MS/MS and HILIC/UPLC(-) MS/MS by UPLC-ESI(-) MS/MS and HILIC UPLC-ESI(-) MS/MS

Line 158: add full stop (“.”) after (19).

Line 164: add full stop at end of sentence

Line 209: remove spurious full stop

Line 316: Next, we assessed...

Line 320: at 32-38 weeks...

Line 323: the 32-38 weeks BMI-adjusted...

Point-by-point Reviewer Response to:

A Metabolomic Signature of Maternal BMI is Associated with Pregnancy Complications: Insights from the COPSAC2010 and VDAART Mother-Child Cohorts

David Horner, MD PhD, Rebecca Vinding, MD PhD, Tingting Wang, PhD, Mina Ali, PhD, Mario Lovric, PhD, Nicole Prince, PhD, Jessica Lasky-Su, ScD, Klaus Bønnelykke, MD PhD, Jakob Stokholm, MD PhD, Bo Chawes, MD PhD DMSc, Morten Arendt Rasmussen, PhD.

This study addresses important research questions and benefits from a rich dataset. However, it would be strengthened by situating itself more explicitly within the existing literature and clearly highlighting its novel contributions.

Reviewer #1:

Reviewer Comment #1

Findings of prior research on the associations between BMI and pregnancy complications should be discussed in the background. Several studies have identified metabolic profiles of BMI and pregnancy complications during pregnancy, including longitudinal associations. Kivelä et al. (cited in the manuscript) have focused on longitudinal associations and identified metabolic changes in pregnancy complications that were independent of BMI. The authors should build their study's justification taking into account this existing evidence, not just dismissing it as inadequate.

Authors Response #1

Thank you for this valuable comment. We agree with the importance of acknowledging prior work on longitudinal metabolic profiling in pregnancy, specifically with regard to focus of our article maternal BMI metabolomic profiling.

In response, we have revised the background section of the introduction to incorporate more detailed findings from the Kivelä et al. which we believe now better contextualises our study. Specifically, we have added the following sentence:

Line 48: *“Moreover, several studies have identified metabolomic alterations associated with pregnancy complications such as pre-eclampsia (9) and gestational diabetes (10), including longitudinal analyses showing that metabolic changes during pregnancy are blunted in women with obesity (5), which may indicate an altered baseline metabolic state.”*

This addition reflects the growing body of evidence on dynamic metabolic patterns in pregnancy and helps clarify how our study builds upon and extends these findings.

Reviewer Comment #2

Another question is clinical relevance: while associations between BMI-related metabolic changes may predict pregnancy complications better when pregnancy complications have already occurred, from the clinical point of view prediction of pregnancy complications is important before their onset.

Authors Response #2

We appreciate this thoughtful comment regarding the timing of prediction in relation to clinical onset. We agree that predictive biomarkers are most valuable when assessed prior to the development of complications. In our study, this temporality is upheld for outcomes such as preeclampsia and caesarean section, where by clinical definition the diagnosis occurred after the timepoint of blood sampling. Thus, for these outcomes, our design ensures that both maternal BMI and metabolomic profiling preceded the complication, supporting their potential utility for early risk stratification.

We acknowledge that for gestational diabetes mellitus, temporality is more challenging to establish definitively, given the structured screening typically conducted during mid-pregnancy. While some GDM cases may have been developing at the time of sampling, we maintain that our modelling remains clinically relevant. Specifically, we modelled BMI-associated metabolomic profiles using a supervised approach based solely on pre-pregnancy BMI, a known risk factor present before conception, enabling us to assess metabolic patterns that may contribute to risk even prior to standard screening windows.

We have clarified these points in the revised manuscript, specifically in the Discussion section, where we now state:

Line 261: *“Previous research has shown that pregnancy-related metabolic changes are smaller in women with obesity, who display metabolic perturbations already in early pregnancy (5). This suggests that the metabolic alterations of obesity and pregnancy disorders share a common origin and supports our observation that higher baseline BMI levels in VDAART enhance the predictive power of metabolite scores, even in early gestation. This interpretation is in line with the concept of pregnancy as a physiological stress test, which may unmask underlying metabolic vulnerability in women predisposed to cardiometabolic disease later in life (22). Moreover research has shown the potential for maternal BMI to moderate the relationship between specific risk-metabolites and pregnancy complications such as preeclampsia and underscores the necessity to account for different maternal risk profiles and associated biomarker profiles (23). Taken together, these findings support the clinical relevance of our approach. While the observational nature of our study limits definitive conclusions about temporality, maternal BMI and metabolomic profiling generally preceded the onset of complications, supporting their potential utility for early risk stratification.”*

Reviewer Comment #3

It is also counterintuitive that the discovery cohort includes only one measurement point, while longitudinal analysis is conducted in a replication cohort – the authors need to justify this approach.

Authors Response #3

We thank the reviewer for this insightful comment. Our study design was shaped by the data availability and distinct strengths of each cohort. The discovery cohort (COPSAC2010) includes a single metabolomic timepoint at 24 weeks' gestation, and offers extensive phenotypic characterisation, which we used to robustly identify maternal BMI-associated metabolites and their associations with pregnancy complications.

The replication cohort (VDAART), in contrast, included serial blood samples from both early (10–18 weeks) and late (32–38 weeks) gestation, enabling a longitudinal exploration of these BMI-associated metabolic profiles. Importantly, training the supervised machine learning model on an entirely independent cohort (COPSAC2010) preserves model independence in the longitudinal replication analyses. Had we instead trained and tested within VDAART across different gestational timepoints, the risk of overfitting to individual-level characteristics would have been substantially higher, compromising the interpretability of temporal comparisons. By deriving the BMI-metabolite model externally, we aimed to avoid this concern and ensured that comparisons between early and late gestational timepoints in VDAART reflect true gestational dynamics rather than within-subject model fitting.

This approach allowed us to (1) validate the generalisability of BMI-metabolite associations across cohorts, and (2) assess how predictive power evolves throughout pregnancy in a way that is both temporally meaningful and methodologically independent. We believe this strategy strengthens the clinical relevance of our findings by simulating real-world scenarios in which a BMI-associated metabolite score could be applied at various stages of pregnancy to inform risk stratification.

Reviewer Comment #4

Statistical methods require more detail. The authors should specify which research question was addressed with each method. Longitudinal analysis is not explained in the methods and it is not clear based on what the conclusions that the associations were stronger towards late gestation were made.

Authors Response #4

We appreciate the reviewer's request for further clarification regarding our statistical methods and the interpretation of gestational timing. The overall aim of our study was not to conduct a longitudinal analysis in the conventional sense, such as modelling within-individual change over time as done by Kivelä et al., but rather to identify metabolic features associated with maternal pre-pregnancy BMI and assess their predictive utility for pregnancy complications. This objective is described at the end of our introduction, where we state:

Line 57: *“By employing untargeted blood metabolomics and supervised machine learning techniques, we aim to elucidate the metabolites and metabolic pathways that underlie these*

associations. We investigate specific metabolites potentially mediating the association between maternal BMI and outcomes where significant associations are observed with both anthropometric measures and metabolite scores. Furthermore, by contrasting BMI-associated metabolite scores at different stages of pregnancy, we explore the dynamic nature of BMI-associated metabolic changes and their significance for predicting maternal health outcomes.”

While we do not claim to perform longitudinal analysis in the traditional sense, we do aim to provide some temporal resolution by comparing the performance of the BMI-associated metabolite scores between early (10–18 weeks) and late (32–38 weeks) gestational timepoints in the VDAART cohort (validation cohort). The observation that associations between metabolite scores and pregnancy complications were stronger at later gestational timepoints is based on the differences in effect estimates and model performance metrics across these two stages. We now clarify this point explicitly in the revised Methods section under ‘Statistical Analysis’, where we state:

Line 485: *“To provide temporal resolution, we applied the COPSAC2010 derived BMI-metabolite scores to VDAART samples at both early (10–18 weeks) and late (32–38 weeks) gestation, allowing us to compare associations across gestational stages and assess whether predictive strength varied by timing..”*

To address the reviewer’s request for clarity on the statistical methods and their respective aims, we have also revised the Methods section under ‘Blood Metabolome’ to specify the purpose of our metabolomic driven analytical approach. In particular, we now clarify:

Line 458: *“We used two VDAART blood metabolome pregnancy time points (10-18 weeks gestation and 32-38 weeks gestation), measured on the same platform as the COPSAC2010 samples, to replicate our findings for pregnancy complications. At the 24-week pregnancy time point in COPSAC2010, we compared overlapping metabolites with those from early pregnancy (10-18 weeks gestation) and late pregnancy (32-38 weeks gestation) in VDAART, identifying 640 and 689 overlapping metabolites, respectively. Using metabolite models trained on COPSAC2010 towards maternal pre-pregnancy BMI, we predicted BMI-metabolite scores in VDAART after applying the same preprocessing steps (log-transformation, centring, and scaling). These scores were then used to assess associations with pregnancy complications in VDAART and to explore potential differences in predictive strength between early and late gestation. This approach allowed for temporal comparison while maintaining methodological independence between training and validation datasets.”*

Reviewer Comment #5

Finally, several studies (including those cited by the authors, as well this Pubmed 35948657) have used similar analytical approaches to identify metabolic profiles of BMI during pregnancy. However, the authors do not discuss how the metabolic profile of BMI identified in their sample align to these previous findings.

Authors Response #5

We thank the reviewer for this constructive comment. We agree that it is important to situate our findings in the context of previous studies that have applied multivariate approaches to define BMI-associated metabolic profiles during pregnancy. While the overarching aim of our study was to investigate associations between maternal BMI, its metabolomic signature, and pregnancy complications, we agree there would be value in providing further discussion reflecting on the metabolic profiling we see in the pregnancy BMI blood metabolome score.

Whilst, many studies such as Girchenko et al. (PMID: 35948657) identified metabolite components associated with maternal early-pregnancy BMI using targeted metabolomics and multivariate methods. Our data-driven BMI-metabolite scores, though derived from untargeted profiling and based on a different population sample, similarly highlighted lipids, amino acids, and sphingolipids as key contributors. This convergence in metabolite classes reinforces the robustness of BMI-associated metabolic disturbances during pregnancy across cohorts and platforms. We have added the incorporated updated text to the discussion to reflect this.

Line 226: *“Several previous studies have used multivariate approaches to define metabolic components associated with maternal BMI, including early pregnancy profiles (6,15,16). Although our study employed an untargeted platform and derived BMI-metabolite scores using external validation, we similarly observed enrichment of lipid-related metabolites, including ceramides and fatty acids, as well as specific amino acid and vitamin metabolism pathways. This overlap supports the reproducibility of BMI-associated metabolic perturbations in pregnancy and highlights their potential generalisability across cohorts and analytical platforms. In our multivariable mediation analysis, we identified 16 metabolites mediating the association between maternal BMI and gestational diabetes. A distinct pattern associated with dietary and lipid metabolism emerged among these 16 metabolites. Ceramide species have been identified in several studies as potential biomarkers for gestational diabetes mellitus. For example, C18:1-Ceramide has been proposed as a potential biomarker for early gestational diabetes detection during the first trimester, demonstrating significant differences between pregnant women with gestational diabetes and those with normal glucose tolerance (17). Sphingolipids, especially ceramides, may contribute to diabetes (18). Accordingly, sphingomyelin profiles in placentas shows significant differences among pregnancies with gestational diabetes (19). Carotene diols from Vitamin A metabolism have been shown to be negatively associated with gestational diabetes, suggesting their potential beneficial effect in reducing gestational diabetes risk. Carotene diol metabolic pathways have been shown as potential mediators, negatively associated with maternal obesity (20), consistent with our findings.”*

Reviewer #2:

In their research Homer et al, used metabolomics data as derived from blood samples taken at 24 weeks of pregnancy from participants of the COPSAC2010 study, to firstly select metabolite to generate a metabolite score to predict pre-pregnancy BMI and then to compare this score with pre-pregnancy BMI in terms of its association with adverse outcomes. In a second step, the researchers applied mediation analysis to refine their metabolite panel and to select a subset of metabolites which potentially indirectly mediate the effect of BMI, an imperfect measure of obesity, on gestational diabetes (GDM), resulting in a second metabolite score. Using Metabolomics data as derived from blood samples at 2 gestational age windows (10-18 wks; 32-38 wks) from a second, independent pregnancy cohort VDAART were used to validate the metabolic scores in a variety of model permutations. The metabolic scores outperformed pre-pregnancy maternal BMI in predicting GDM and pre-eclampsia, with indications that these scores can be used to improve the prediction of adverse pregnancy outcomes when combined with pre-pregnancy BMI. This suggest that the total effect of BMI on pregnancy outcome has a direct and an indirect effect component. The authors also find that the metabolic scores prediction ability increase with gestational age.

RESEARCH IMPACT:

The research of Horner et al. add to the concept that BMI is an imperfect measure of obesity, and therefore maternal pre-pregnancy BMI is a limited predictor of adverse pregnancy outcomes. With their research Horner et al show that specific panel of metabolites have the ability to complement pre-pregnancy BMI, possibly through a better capture of obesity, and thus improve prediction of pregnancy outcomes.

Reviewer Comment #6

1.Abstract: Some suggestion to improve readability -

1.1: Lines 14-15: Consider simplifying the sentence: [The rising incidence of] Maternal obesity is linked to pregnancy complications, including gestational diabetes, preeclampsia, and caesarean section [, driven by underlying metabolic disturbances]. -

Authors Response #6

Thank you for your introductory comments regarding the general themes of our manuscript. We agree that maternal BMI is an imperfect proxy for metabolic health and appreciate the reviewer's support for our use of metabolomics to refine risk assessment.

Regarding the specific suggestion to simplify our abstract, we agree that a more concise formulation enhances readability and impact in this context. We believe the revised version improves flow and sharpens the focus on the core message. The updated sentence is now:

Line 14: *"Maternal obesity is increasingly common and linked to pregnancy complications, likely driven by underlying metabolic perturbations."*

Reviewer Comment #7

1.2: Lines 17-18: Use of abbreviations is not advised in an abstract: Spell out BMI and the cohort study acronyms COPSAC2010 and VDAART at first occurrence -

Authors Response #7

Thank you for this helpful suggestion. We agree that abbreviations should be avoided in the abstract where possible. We have revised the abstract text to spell out both the cohort acronyms and "BMI" at first mention to improve accessibility and readability.

Reviewer Comment #8

1.3: Line 18-19: Consider rephrasing: "with untargeted blood metabolomics sampling taken during early-, mid- and late gestation" as follows "with untargeted blood metabolomics performed on blood samples taken during early-, mid- and late gestation" -

Authors Response #8

Thank you for this suggestion. We agree that the proposed phrasing improves clarity and precision. We have updated the sentence accordingly.

Line 17: *"Data from the Copenhagen Prospective Studies on Asthma in Childhood 2010 (COPSAC2010) and Vitamin D Antenatal Asthma Reduction Trial (VDAART) cohorts were used, with untargeted blood metabolomics performed on blood samples taken during early-, mid-, and late gestation."*

Reviewer Comment #9

1.4: Line 26-27: Consider rephrasing: "Mediation analysis in COPSAC2010 identified 16 metabolites that mediated the BMI-gestational 26 diabetes association." As follows "Mediation analysis in COPSAC2010 identified 16 metabolites as mediating the effect of BMI on gestational diabetes" -

Authors Response #9

Thank you for this suggestion. We agree that the revised phrasing improves clarity and flow. The sentence has been updated accordingly.

Line 26: "Mediation analysis in COPSAC2010 identified 16 metabolites as mediating the effect of BMI on gestational diabetes."

Reviewer Comment #10

1.5: Lines 30-31: Remove the following sentence section: "offering potential for targeted interventions" [There is no basis in the paper to support this forward-looking possibility]

Authors Response #10

Thank you for the comment. We have removed the phrase as suggested. We would highlight this was intended as a forward-looking perspective for future work, and the original language reflected this rather than a definitive claim.

Reviewer Comment #11

2. Introduction: -

2.1: Lines 36-39: Reformulate the sentences starting with "It is unlikely that... and child birth outcomes" to reflect that BMI is a simple anthropometric measure widely used to assess adiposity and thus obesity. To overcome the limitations of BMI in identifying obesity, the use of metabolomics has been proposed to redefine BMI in terms of metabolic health; the authors may want to refer to Ottoson et al. to make this point; doi.org/10.2337/dc21-2402. By reframing the research in this fashion, the added value of the research will become clearer.

Authors Response #11

Thank you for this helpful suggestion. We have revised the passage to reflect the limitations of BMI as a proxy for metabolic health and to clarify the motivation for a metabolomics-based approach. We have also cited the recommended reference to support this perspective.

Line 36: *"Body mass index (BMI) is a widely used anthropometric measure of adiposity, but it does not capture the underlying metabolic heterogeneity associated with obesity. Recent studies have shown that individuals with a normal BMI can exhibit an obesity-related metabolome and elevated risk of cardiometabolic disease, suggesting that BMI alone may underestimate metabolic risk (4). These perturbations, reflected in BMI-associated metabolites, have thus been linked to adverse pregnancy (5) and child birth outcomes (6)."*

Reviewer Comment #12

- 2.2: Line 37 and throughout manuscript: Consider use of "Perturbation" rather than "Disturbance"

Authors Response #12

We agree with this suggestion and have replaced all instances of "disturbance" with "perturbation" throughout the manuscript to reflect the appropriate terminology.

Reviewer Comment #13

- 2.3: Line 48: Reference #9 is not relevant the context of metabolites for the prediction of preeclampsia. Instead, the authors should consider referring McBride et al; doi.org/10.3390/metabo11080530. McBride et al have performed metabolomics studies across different pregnancy cohorts evaluating prediction of different pregnancy outcomes akin the research of Horner & colleagues. This study also uses the Metabolon metabolomics analysis pipeline and reports several of the metabolites identified in the 46 marker panel, providing cues for the discussion of findings by Horner & colleagues.

Authors Response #13

We thank the reviewer for this valuable suggestion. We have replaced the previous reference with the recommended citation (McBride et al., Metabolites 2021) in the Introduction to better reflect the relevant literature on metabolomics-based prediction of pregnancy complications. In addition, we have incorporated discussion of this study in the Discussion section to highlight the consistency of findings across cohorts using the same metabolomics platform. Specifically, we now note that glycerol and 4-hydroxyglutamate, two top predictors in McBride et al's work, were also retained in our maternal BMI model and significantly mediated the association between maternal BMI and gestational diabetes in our analysis.

Line 253: *“Our work aligns with previous work (9), where the addition of metabolites, measured using the same platform, improved prediction of pregnancy-related disorders, including gestational diabetes and preeclampsia, beyond established risk factors. Notably, glycerol and 4-hydroxyglutamate were among the top predictors for these outcomes and were also retained in our maternal BMI model. In our analysis, both metabolites significantly mediated the association between maternal BMI and gestational diabetes.”*

Reviewer Comment #14

- 2.4: Line 54: Consider rephrasing: “...to investigate the relationship between maternal BMI and pregnancy complications through blood metabolomics” as follows “...to investigate the relationship between maternal BMI, obesity and pregnancy 54 complications through blood metabolomics

Authors Response #14

Thank you for the suggestion. We agree that explicitly mentioning obesity helps clarify the scope of the investigation. We have rephrased the sentence accordingly.

Line 55: “To address these gaps, our study leverages the large and well-characterised mother-child cohorts, COPSAC2010 and VDAART, to investigate the relationship between maternal BMI, obesity, and pregnancy complications through blood metabolomics.”

Reviewer Comment #15

3. Results:

- Line 65: Rephrase: “... 684 mothers had untargeted blood metabolome sampling at 24 weeks of gestation.” as follows “... 684 has untargeted metabolomics perform on blood samples taken at 24 weeks of gestation.”

Authors Response #15

Thank you for this suggestion. We have changed the text accordingly.

Line 66: “In the COPSAC2010 cohort, we included 690 mothers with data on maternal pre-pregnancy BMI. 684 mothers had untargeted metabolomics performed on blood samples taken at 24 weeks of gestation.”

Reviewer Comment #16

- Line 66: The authors should consider stratifying the patients into the well established BMI classes, non- overweight BMI<25, overweight 25 == 30, this will facilitate the interpretability of the baseline characteristics of the study population. It is suggested to add stratification in BMI classes as additional levels of information to the characteristic “Maternal Pre-pregnancy BMI.

Authors Response #16

We agree that stratifying by standard clinical BMI categories provides important context. Accordingly, we have included a new supplementary table (Table S1) presenting baseline characteristics across maternal BMI categories: underweight (<18.5), normal weight (18–25), overweight (25–30), and obesity (>30), following standard definitions. Notably, we found no substantial differences in baseline characteristics across these strata.

Due to group size imbalances and our interest in capturing potential linear associations with maternal BMI, we have retained the tertile-based stratification in the main table 1. We believe this approach better reflects the continuous nature of BMI associations in our cohort.

Revision in main text:

Line 71: *“Baseline characteristic differences were similar across strata of clinical obesity categories (Table S1).”*

New Supplementary Table added:

(see next page)

Baseline Characteristics	<18 (Underweight)	18–25 (Normal)	25–30 (Overweight)	>30 (Obese)	p-value
n =	8	436	172	74	
Male Sex (%)	3 (37.5)	224 (51.4)	92 (53.5)	34 (45.9)	0.618
Caucasian Race (%)	8 (100.0)	418 (95.9)	167 (97.1)	69 (93.2)	0.51
Income type (%)					0.848
Low	0 (0.0)	37 (14.3)	14 (12.4)	9 (15.5)	
Medium	2 (50.0)	74 (28.7)	39 (34.5)	16 (27.6)	
High	2 (50.0)	147 (57.0)	60 (53.1)	33 (56.9)	
Maternal Education Level at Birth					<0.001
Low	0 (0.0)	22 (5.0)	18 (10.5)	10 (13.5)	
Medium	5 (62.5)	267 (61.2)	107 (62.2)	59 (79.7)	
High	3 (37.5)	147 (33.7)	47 (27.3)	5 (6.8)	
Maternal age at birth (mean (SD))	31.27 (4.17)	32.31 (4.23)	32.18 (4.61)	32.25 (4.53)	0.909
Birthweight (mean (SD))	3.16 (0.48)	3.50 (0.55)	3.64 (0.52)	3.65 (0.48)	0.001
Gestational age (mean (SD))	273.38 (14.13)	278.79 (11.70)	280.51 (11.49)	279.47 (8.62)	0.175
Cesarean section (%)	1 (12.5)	83 (19.0)	43 (25.0)	21 (28.4)	0.152
Maternal smoking during pregnancy (%)	1 (12.5)	31 (7.1)	16 (9.3)	6 (8.1)	0.782
Siblings (mean (SD))	1.25 (0.46)	1.54 (0.78)	1.57 (0.79)	1.69 (1.34)	0.441
Pregnancy Varied Dietary Pattern (mean (SD))	0.59 (0.78)	0.07 (0.99)	-0.11 (0.93)	-0.05 (1.20)	0.135
Pregnancy Western Dietary Pattern (mean (SD))	0.53 (0.98)	-0.08 (1.02)	0.01 (0.88)	0.30 (1.06)	0.02
Pregnancy PC3 Dietary Pattern (mean (SD))	0.29 (0.50)	0.08 (1.00)	-0.04 (0.98)	-0.33 (0.98)	0.02
Maternal Pre-pregnancy BMI (mean (SD))	16.99 (0.81)	22.11 (1.66)	27.00 (1.42)	34.05 (3.72)	<0.001
Maternal BMI Metabolite Score (mean (SD))	-0.19 (0.53)	-0.44 (0.79)	0.40 (0.80)	1.39 (0.81)	<0.001

Table S1. Baseline characteristics stratified by clinical maternal pre-pregnancy BMI categories (underweight, normal weight, overweight, and obese) among pregnant mothers in the COPSAC2010 cohort.

Reviewer Comment #17

- Lines 72-76: Can the authors clarify whether the BMI prediction considered only women with uncomplicated pregnancy outcomes (to avoid confounding by the adverse outcomes) or were samples of all women used in the prediction. The following constitutes a preferred data analysis presentation: - develop BMI predictor based on metabolite profiles in blood samples of uncomplicated pregnancies - plot the metabolite predicted BMI versus the measured pre-pregnancy BMI for the uncomplicated pregnancies - plot on top, using different colours/symbols, study participants with the various pregnancy complications; as such one can visually gauge the added value of metabolite predicted BMI over measured pre-pregnancy BMI.

Authors Response #17

We thank the reviewer for this thoughtful and constructive suggestion. In our current analysis, we included all women in the COPSAC2010 cohort, regardless of pregnancy outcome, when training the BMI prediction model using supervised sparse partial least squares regression. This approach was chosen to maximise statistical power and ensure model stability, particularly since adverse pregnancy outcomes represent a minority of the cohort. Our goal was to model general BMI-related metabolic profiles, rather than outcome-specific variation.

We did initially consider restricting the model training to women with uncomplicated pregnancies. However, doing so would have resulted in a fundamentally different model, as the sparsity constraints inherent to our approach would likely have selected a different set of predictive metabolites. This would limit interpretability and complicate integration with the rest of our manuscript findings where other models are utilised.

Nonetheless, we fully agree that visualisation of the relationship between metabolite-predicted BMI and pregnancy complications is valuable. To address this, we have added a new supplementary figure (Figure S1), where the relationship between metabolite-predicted BMI and measured pre-pregnancy BMI is shown for all women. Women without complications are displayed in grey, while those with specific complications are highlighted in color, with ellipses indicating the variation within each group. This visualization enables readers to intuitively assess whether women with complications deviate from the general trend. We believe this figure provides an informative and interpretable complement to our main analysis.

We have therefore included the following additional text in our manuscript:

Line 79: “Further, to evaluate whether the relationship between predicted and measured BMI varied by pregnancy complication status, we visualised this association across maternal complication groups. Figure S1 shows the relationship between maternal BMI metabolite score and measured pre-pregnancy BMI, with women experiencing specific complications highlighted against the broader cohort.”

BMI Prediction vs Measured BMI by Pregnancy Complication

Grey: No complication; Colored: Specific complication group

Figure S1. Relationship between Metabolite-Predicted and Measured Pre-Pregnancy BMI by Pregnancy Complication Status in COPSAC2010.

Reviewer Comment #18

- Line 76: Replace “employed” by “performed”

Authors Response #18

We have made this change.

Line 83: *“We performed pathway enrichment analysis on these 46 metabolites, considering 25 sub pathways.”*

Reviewer Comment #19

- Lines 76 – 79: The recent comment in Nature Metabolism provided a reminder on the importance of providing sufficient detail on how pathway enrichment analysis is performed (<https://doi.org/10.1038/s42255-025-01283-0>). There is no information in the paper about the methodology used to perform pathway enrichment analysis. Good practice reporting guidance can f.i. be found in <https://doi.org/10.1021/acs.est.2c05588>. The authors should provide sufficient detail on the pathway analysis performed,

Authors Response #19

We thank the reviewer for this important comment and fully agree that pathway enrichment analysis should be transparently and reproducibly reported. We acknowledge that it was an oversight not to include specific methodological details for this analysis in the original submission, and we appreciate the opportunity to correct this.

To address this, we have now added a description of the methodology used to perform pathway enrichment analysis in the Methods section of the manuscript.

Line 499: *“We performed pathway enrichment analysis to identify overrepresented metabolic pathways among the 46 BMI-associated metabolites selected by sparse partial least squares regression. Annotation was based on the Metabolon-provided biochemical classifications, linking each metabolite to its respective sub- and super-pathways. Enrichment was assessed using hypergeometric testing, comparing the observed count of selected metabolites per sub-pathway to the expected count based on all measured metabolites. Fold enrichment and p-values were computed, with false discovery rate and Bonferroni correction applied to account for multiple testing.”*

Reviewer Comment #20

- Line 79 – 83: Did the authors observe specific missingness patterns in the xenobiotic metabolites?

Authors Response #20

We thank the reviewer for this insightful question. Our primary focus was on the robustness of the metabolite-predicted BMI model rather than pathway-specific differences in missingness. As such, we did not stratify missingness analyses by individual super- or sub-pathways (e.g., xenobiotics), and this was not a primary focus of the study. However, to

address the reviewer's request, we have provided the missingness values for the subset of metabolites classified under the "Xenobiotics" super-pathway below for their reference (See Reviewer file attached to our submission). As perhaps expected, we find that xenobiotic metabolites were among the pathways with the highest proportion of missing values. This aligns with expectations given that these compounds often reflect exogenous exposures (e.g., diet, medications) and are not consistently detected across individuals. To mitigate bias from missingness patterns, we excluded metabolites with high levels of missingness (>33%) prior to imputation and subsequent modeling, as described in our Methods section.

Reviewer Comment #21

- Line 84: The authors suggest that their imputation approach considers the non-randomness of missing data; there is no further explanation to substantiate this claim. The authors are invited to provide some further context.

Authors Response #21

We appreciate the reviewer's suggestion and agree that further explanation is warranted. In our analysis, we excluded metabolites with >33% missingness to avoid unstable imputation. For the remaining data, we applied random forest (RF) imputation using the missforest R package, which models complex, non-linear relationships among metabolites and does not assume that missingness is random.

RF imputation is particularly suited for metabolomics data, where missingness may reflect values below the detection limit or biological variation. As demonstrated by Koka et al (PMID 31601178) and Wei et al (PMID 29330539), RF outperforms minimum value imputation in preserving data structure, reducing bias, and improving downstream analyses. We have revised the methods section to clarify this point and provide appropriate referencing.

Line 452: *"Data preprocessing involved excluding metabolites with more than 33% missing values. For the remaining missing data, random forest imputation was applied using the missForest R package (v1.5) (29). This method does not assume data are missing at random and has been shown to outperform minimum value imputation in preserving the structure and variability of metabolomics data (30)."*

Reviewer Comment #22

- Lines 88 – 95: Assuming that all reported odds ratios in this paragraph are adjusted odds ratios, the readability of this paragraph can be improved by moving the sentence regarding the covariates which were considered in multivariable modelling forward in the paragraph.

Authors Response #22

We thank the reviewer for this helpful suggestion to improve clarity. We have now moved the information on covariate adjustment to the beginning of the paragraph to immediately indicate that the reported odds ratios are from multivariable models.

Line 95: *"Maternal pre-pregnancy BMI was significantly associated with several pregnancy complications in multivariable models adjusted for social circumstances, child sex, maternal smoking, and maternal dietary patterns (Table 2). Per standard deviation (SD) increase in*

BMI, the odds of gestational diabetes increased (OR 1.90 [1.29–2.74], $p < 0.001$), as did the odds of caesarean section (OR 1.23 [1.03–1.47], $p = 0.023$), induction of birth (OR 1.42 [1.21–1.67], $p < 0.001$), and maternal antibiotics at birth (OR 1.18 [1.01–1.39], $p = 0.042$).

Reviewer Comment #23

- Lines 96 -103: This paragraph will benefit from data presentation as outlined in comments "lines 72-76"

Authors Response #23

We thank the reviewer for this helpful suggestion. To improve clarity and consistency, we have revised the paragraph to explicitly reference the metabolite score derived via sparse partial least square modelling described in lines 72–76, and have aligned the structure and presentation of the results accordingly. The updated text also references the relevant figure and table showing metabolite loadings and model performance.

Line 104: *"The maternal BMI metabolite score, derived using sparse partial least squares modelling as described above (Figure 1, Table S3), was significantly associated with several pregnancy complications. Per SD increase in the score, the odds of gestational diabetes increased (OR 2.47 [1.45–4.24], $p < 0.001$), as did the odds of caesarean section (OR 1.29 [1.06–1.58], $p = 0.011$), induction of birth (OR 1.31 [1.10–1.56], $p = 0.002$), and maternal antibiotics at birth (OR 1.21 [1.01–1.44], $p = 0.035$), with slightly higher estimates than for maternal BMI alone (Table 2)."*

Reviewer Comment #24

- Lines 105 115: - Reference to Table S1: The authors should consider stratifying the patients into the well established BMI classes, non- overweight BMI < 25, overweight 25 == 30, this will facilitate the interpretability of the baseline characteristics of the study population. It is suggested to add stratification in BMI classes as additional levels of information to the characteristic "Maternal Pre-pregnancy BMI. Distributing the outcomes over the different BMI classes may further add to the interpretability (<> small numbers / class).

Authors Response #24

We thank the reviewer for this valuable suggestion. We have now included BMI category stratifications in Table S1 using standard World Health Organization (WHO) categories: underweight (<18), normal weight (18–25), overweight (25–30), and obese (>30). This provides greater granularity to the maternal BMI characteristic and enhances comparison across the two cohorts. The updated Table S1 now includes these categorical BMI distributions for both the COPSAC2010 and VDAART cohorts, with corresponding p-values. This addition improves interpretability by clearly illustrating the differences in maternal weight status across cohorts.

Cohort Characteristics	COPSAC2010	VDAART	p-value
n =	684	775	
Male Sex (%)	350 (51.2)	402 (51.9)	0.81
Caucasian Race (%)	655 (95.8)	304 (40.9)	<0.001
Income type (%)			<0.001
Low	58 (13.6)	97 (35.8)	
Medium	128 (30.0)	93 (34.3)	
High	240 (56.3)	81 (29.9)	
Maternal Education Attainment at Birth			<0.001
Low	50 (7.3)	285 (38.4)	
Medium	434 (63.5)	204 (27.5)	
High	200 (29.2)	254 (34.2)	
Maternal age at birth (years) (mean (SD))	32.30 (4.36)	25.39 (5.47)	<0.001
Maternal Pre-pregnancy BMI (mean(SD))	24.56 (4.41)	28.38 (7.79)	<0.001
Maternal Pre-pregnancy BMI Obesity Category			
<18 (Underweight)	8 (1.2)	11 (1.7)	<0.001
18–25 (Normal)	428 (63.0)	235 (36.8)	
25–30 (Overweight)	170 (25.0)	188 (29.4)	
>30 (Obese)	73 (10.8)	205 (32.1)	
Birthweight (kilograms) (mean (SD))	3.55 (0.54)	3.27 (0.57)	<0.001
Gestational age (days) (mean (SD))	279.28 (11.30)	272.96 (13.85)	<0.001
Maternal smoking during pregnancy (%)	52 (7.6)	18 (2.3)	<0.001
Gestational Diabetes (%)	15 (2.2)	40 (5.2)	0.005
Preeclampsia (%)	29 (4.2)	34 (4.4)	0.981
C-section (%)	148 (21.6)	229 (29.5)	0.001
Acute C-section (%)	83 (12.1)	70 (9.0)	0.065
Elective C-section (%)	65 (9.5)	159 (20.5)	<0.001
Induction of Birth (%)	243 (35.6)	141 (18.2)	<0.001
Maternal Antibiotics at Birth (%)	219 (32.2)	299 (38.7)	0.011

Reviewer Comment #25

Reference to Supplementary Figure 2: It appears to the reviewer that the left top panels show that the metabolite profiles are quite distinct between COPSAC 24 wks and VDAART 10-18 wks; yet that the difference is smaller when restricting the data set to the selection of the 46 metabolites. The right top panels appear to indicate that the COPSAC 24 wks and VDAART 32-38 wks metabolite profiles are less different and that this difference is even less

when restricting the data set to the selection of the 46 metabolites. Is this interpretation correct; if yes, it may be worthwhile for the authors to comment on this. The same holds true for the bottom panels of Figure S2 – elaboration on how to interpret (46 metabolite panels are less prone to time cohort differences than “all metabolites”) and what this means for the study results would add to the interpretability of the 46 metabolite selection.

Authors Response #25

We thank the reviewer for this interpretation and valuable suggestion. As requested, we have now clarified and elaborated on the interpretation of Supplementary Figure 3 (formerly Supplementary Figure 2), in the figure legend.

Specifically, the PCA plots demonstrate that the maternal metabolomic profiles from COPSAC2010 (24 weeks gestation) and VDAART (10–18 weeks and 32–38 weeks gestation) are more distinct when all available metabolites are used, compared to when only the 46 score metabolites are included. This is expected, as the full metabolome captures a broad array of both biological and technical variability.

Importantly, the metabolite scores used in our analyses are directly comparable across cohorts because they are derived from the same subset of metabolites using identical preprocessing and imputation workflows). Our modeling operates in relative space, where both cohorts are independently mean-centered and scaled prior to analysis. Therefore, the purpose of the comparison is not to align absolute metabolite levels or distributions, but rather to ensure that the variance structure of the selected 46 metabolites is sufficiently similar across cohorts. This comparable variance is what provides utility and validity for applying the model across independent datasets.

Updated Figure Text:

Figure S3. The figure compares maternal blood metabolomes at three different pregnancy timepoints from two mother-child cohorts, COPSAC2010 and VDAART. The top panels display a Principal Component Analysis (PCA) score plot for all metabolites and the selected metabolite scores comparing COPSAC2010 vs VDAART at 10-18 weeks (left) and 32-38 weeks (right) of gestation. The bottom panels illustrate the relative variation per metabolite, computed as the ratio of sums of squares ($SSQ_{\text{time}} / SSQ_{\text{residual}}$) from a one-way ANOVA model with Time/Cohort as the predictor. Additionally, it compares the per metabolite standard deviation within the cohort relative to the 24-week gestation time point from COPSAC2010, shown for COPSAC2010 vs VDAART at 10-18 weeks (left) and 32-38 weeks (right). Importantly, as modeling operates in relative space with independent centering and scaling per cohort, this comparison ensures stable variance structure of the 46 selected metabolites rather than alignment of absolute levels.

Reviewer Comment #26

- Lines 117 – 128: To improve readability, consider starting with describing the pre-pregnancy BMI associations with pregnancy outcomes in VDAART (cf lines 124 -128) and then to expand to validating the COPSAC 24 weeks BMI associated metabolite models in the 2 VDAART metabolome time points (cf lines 17 -123). Once again, plotting the measured BMI vs the metabolite BMI scores and colour/symbol coding the various pregnancy outcomes are likely to convey the added value of the metabolite BMI scores better.

Authors Response #26

Thank you for this helpful suggestion. We have revised the structure of this section to improve clarity by first presenting the associations between maternal pre-pregnancy BMI and pregnancy outcomes in the VDAART cohort. We then describe how the metabolite-based BMI scores, trained in COPSAC2010, replicate these associations at both early and late pregnancy timepoints in VDAART.

Line 127: *“Maternal pre-pregnancy BMI was significantly associated with several pregnancy complications in the VDAART cohort in multivariable models adjusted for social circumstances, child sex, and maternal smoking (Table 2). Per standard deviation (SD) increase in BMI, the odds of gestational diabetes increased (OR 1.49 [1.08–2.02], $p=0.012$), as did the odds of caesarean section (OR 1.36 [1.14–1.61], $p<0.001$) and induction of birth (OR 1.30 [1.07–1.57], $p=0.008$), whereas no significant associations were observed for preeclampsia (OR 1.11 [0.76–1.55], $p=0.553$) or maternal antibiotics at birth (OR 1.11 [0.94–1.31], $p=0.196$).”*

To test the generalisability of our metabolite-based BMI models, we applied the 46-metabolite COPSAC2010-trained prediction scores to VDAART maternal blood samples at the gestational windows: early (10–18 weeks) and late pregnancy (32–38 weeks). These scores were strongly associated with actual maternal pre-pregnancy BMI (10–18 weeks: β 4.62, $p=10^{-63}$; 32–38 weeks: β 4.26, $p=10^{-54}$), validating the metabolite signature’s consistency across gestation and independent cohorts. Figure S4 illustrates the relationship between the maternal BMI metabolite score and measured pre-pregnancy BMI in the VDAART cohort, with women experiencing specific pregnancy complications highlighted against the broader cohort.”

As mentioned above, we have now added a visual validation of the metabolite-predicted BMI scores against measured maternal BMI, stratified by pregnancy complications in the VDAART cohort (see Supplementary Figure 4). Each subplot represents a specific complication with "Yes" cases color-coded and "No" cases shown in grey. This visualisation confirms that the metabolite scores reliably track with measured BMI, while also illustrating how the scores distribute across relevant pregnancy outcomes.

(Figure provided in next page)

VDAART BMI Prediction vs Measured BMI by Pregnancy Complication

Grey: No complication; Colored: Specific complication group

Figure S1. Relationship between Metabolite-Predicted and Measured Pre-Pregnancy BMI by Pregnancy Complication Status in VDAART

Reviewer Comment #27

- Lines 129 – 134: this paragraph contains some of the key results of the manuscript and the authors should consider expanding this section. For instance, did the authors attempt delineating the metabolites in function of the outcomes GDM and PE?

Authors Response #27

We thank the reviewer for this insightful suggestion. In response, we have now conducted additional multivariable analyses to assess associations between each of the 46 maternal BMI-associated metabolites and the risk of gestational diabetes (GDM) and preeclampsia (PE), separately at early (10–18 weeks) and late (32–38 weeks) pregnancy. These analyses were adjusted for the same multivariable analysis as the main results.

We have added the following sentence to the main text to reflect this addition

Line 146: *"To identify specific metabolites contributing to these associations, we examined associations between each of the 46 BMI-related metabolites at each timepoint and the risk of gestational diabetes and preeclampsia at early and late pregnancy (Table S6)."*

These findings provide additional insights into BMI associated metabolite-specific contributions to GDM and PE risk, which may be of interest to some readers, and further strengthen the interpretation of our validation analyses.

See new Supplementary Table below

Maternal BMI Associated Metabolites in VDAART	Pre-eclampsia		Gestational Diabetes	
	10-18 Weeks	32-38 Weeks	10-18 Weeks	32-38 Weeks
16a-hydroxy DHEA 3-sulfate	1.02 [0.72 - 1.46], p = 0.900	1.16 [0.8 - 1.67], p = 0.434	1.36 [0.97 - 1.91], p = 0.073	1.49 [1.07 - 2.08], p = 0.019
1-lignoceroyl-GPC (24:0)	-	1.05 [0.72 - 1.52], p = 0.802	-	0.45 [0.32 - 0.63], p = <0.001
1-palmitoleoyl-GPC* (16:1)*	1.23 [0.86 - 1.77], p = 0.256	1.11 [0.77 - 1.61], p = 0.577	0.96 [0.69 - 1.36], p = 0.835	0.58 [0.41 - 0.81], p = 0.001
1-palmitoyl-2-arachidonoyl-GPC (16:0/20:4n6)	1.08 [0.76 - 1.55], p = 0.654	0.91 [0.63 - 1.34], p = 0.642	1.2 [0.87 - 1.66], p = 0.283	1.21 [0.88 - 1.68], p = 0.258
1-palmitoyl-2-arachidonoyl-GPE (16:0/20:4)*	1.38 [0.97 - 1.98], p = 0.079	1.45 [1 - 2.14], p = 0.054	1.28 [0.92 - 1.79], p = 0.144	1.65 [1.17 - 2.36], p = 0.005
1-stearoyl-2-arachidonoyl-GPC (18:0/20:4)	0.9 [0.63 - 1.3], p = 0.559	0.96 [0.66 - 1.43], p = 0.849	1.07 [0.78 - 1.49], p = 0.671	1.26 [0.9 - 1.79], p = 0.178
1-stearoyl-2-arachidonoyl-GPE (18:0/20:4)	1.28 [0.89 - 1.86], p = 0.194	1.13 [0.78 - 1.68], p = 0.524	1.41 [1.01 - 2], p = 0.051	2.49 [1.72 - 3.68], p = <0.001
2,6-dihydroxybenzoic acid	0.85 [0.58 - 1.24], p = 0.395	1.03 [0.67 - 1.57], p = 0.907	0.97 [0.67 - 1.41], p = 0.878	0.81 [0.54 - 1.22], p = 0.318
3beta-hydroxy-5-cholestenoate	1.06 [0.75 - 1.5], p = 0.735	0.99 [0.68 - 1.44], p = 0.962	0.53 [0.37 - 0.73], p = <0.001	0.66 [0.47 - 0.92], p = 0.015
3-phenylpropionate (hydrocinnamate)	1.07 [0.74 - 1.56], p = 0.734	0.88 [0.6 - 1.29], p = 0.496	0.89 [0.65 - 1.22], p = 0.456	0.72 [0.53 - 1], p = 0.045
4-ethylphenyl sulfate	0.8 [0.54 - 1.16], p = 0.261	1.11 [0.75 - 1.61], p = 0.592	1.08 [0.78 - 1.46], p = 0.635	1 [0.73 - 1.35], p = 0.988
4-hydroxyglutamate	1.17 [0.83 - 1.65], p = 0.377	2.05 [1.41 - 3.01], p = <0.001	2.01 [1.43 - 2.82], p = <0.001	1.75 [1.25 - 2.48], p = 0.001
aconitate [cis or trans]	1.1 [0.75 - 1.77], p = 0.676	1.49 [0.94 - 2.44], p = 0.109	1.75 [1.1 - 2.84], p = 0.020	1.88 [1.23 - 2.91], p = 0.004
adipoylcarnitine (C6-DC)	1.32 [0.95 - 1.84], p = 0.097	1.51 [1.06 - 2.14], p = 0.020	0.98 [0.7 - 1.36], p = 0.889	1.8 [1.33 - 2.45], p = <0.001
alpha-ketoglutarate	1.05 [0.76 - 1.49], p = 0.770	1.11 [0.77 - 1.56], p = 0.570	1.02 [0.74 - 1.45], p = 0.910	0.98 [0.7 - 1.37], p = 0.922

andro steroid monosulfate C ₁₉ H ₂₈ O ₆ S (1)*	1.1 [0.77 - 1.57], p = 0.594	1.36 [0.94 - 2], p = 0.107	1.28 [0.91 - 1.79], p = 0.157	1.32 [0.93 - 1.86], p = 0.117
androstenediol (3beta,17beta) disulfate (1)	1.1 [0.78 - 1.57], p = 0.581	1.25 [0.86 - 1.83], p = 0.254	2.02 [1.43 - 2.89], p = <0.001	1.17 [0.84 - 1.63], p = 0.363
androstenediol (3beta,17beta) disulfate (2)	1.09 [0.77 - 1.54], p = 0.617	1.07 [0.74 - 1.56], p = 0.716	1.57 [1.14 - 2.2], p = 0.005	1.23 [0.89 - 1.7], p = 0.199
asparagine	0.93 [0.75 - 1.32], p = 0.604	0.74 [0.49 - 1.09], p = 0.131	0.95 [0.75 - 1.37], p = 0.724	0.58 [0.4 - 0.82], p = 0.003
aspartate	1.03 [0.71 - 1.46], p = 0.882	1.17 [0.8 - 1.67], p = 0.403	1.14 [0.83 - 1.57], p = 0.409	1.5 [1.09 - 2.06], p = 0.012
beta-cryptoxanthin	0.78 [0.57 - 1.1], p = 0.149	0.66 [0.46 - 0.97], p = 0.033	0.64 [0.48 - 0.87], p = 0.003	0.64 [0.47 - 0.88], p = 0.006
branched chain 14:0 dicarboxylic acid	0.97 [0.68 - 1.4], p = 0.875	0.87 [0.58 - 1.28], p = 0.486	0.87 [0.62 - 1.21], p = 0.395	0.71 [0.5 - 1], p = 0.055
carotene diol (1)	0.79 [0.57 - 1.12], p = 0.163	0.64 [0.46 - 0.92], p = 0.011	0.91 [0.67 - 1.26], p = 0.549	0.96 [0.7 - 1.34], p = 0.811
carotene diol (2)	0.86 [0.62 - 1.23], p = 0.412	0.71 [0.5 - 1.03], p = 0.066	0.91 [0.67 - 1.24], p = 0.535	0.88 [0.65 - 1.21], p = 0.429
ceramide (d18:2/24:1, d18:1/24:2)*	0.99 [0.69 - 1.4], p = 0.951	-	0.84 [0.61 - 1.16], p = 0.281	-
cortolone glucuronide (1)	1.24 [0.88 - 1.79], p = 0.236	1.24 [0.86 - 1.82], p = 0.255	1.7 [1.19 - 2.48], p = 0.004	1.36 [0.97 - 1.93], p = 0.082
cys-gly, oxidized	1.04 [0.74 - 1.44], p = 0.808	1.13 [0.8 - 1.55], p = 0.478	0.93 [0.66 - 1.29], p = 0.693	1.01 [0.71 - 1.42], p = 0.933
cysteinylglycine disulfide*	1.05 [0.78 - 1.54], p = 0.788	1.31 [0.93 - 1.8], p = 0.113	1.25 [0.88 - 1.78], p = 0.220	1.22 [0.87 - 1.68], p = 0.248
ergothioneine	0.85 [0.59 - 1.23], p = 0.382	0.99 [0.68 - 1.46], p = 0.940	1.22 [0.86 - 1.72], p = 0.263	1.31 [0.9 - 1.91], p = 0.161
gentisate	0.87 [0.6 - 1.24], p = 0.438	1.1 [0.75 - 1.6], p = 0.613	1.08 [0.78 - 1.48], p = 0.631	0.63 [0.44 - 0.89], p = 0.009
glutamate	1.01 [0.71 - 1.41], p = 0.957	1.24 [0.86 - 1.79], p = 0.240	1.25 [0.91 - 1.69], p = 0.153	1.51 [1.09 - 2.12], p = 0.015
glycerol	1.24 [0.87 - 1.81], p = 0.254	1.42 [0.95 - 2.19], p = 0.105	1.26 [0.9 - 1.8], p = 0.185	1.99 [1.33 - 3.07], p = 0.001
glycosyl-N-(2-hydroxynervonoyl)-sphingosine (d18:1/24:1(2OH))*	1.11 [0.79 - 1.58], p = 0.543	1.15 [0.8 - 1.73], p = 0.488	1.16 [0.83 - 1.65], p = 0.389	1.45 [0.99 - 2.23], p = 0.076
hydroxyasparagine	1.22 [0.87 - 1.73], p = 0.248	2.14 [1.53 - 3], p = <0.001	1.26 [0.9 - 1.77], p = 0.178	1.15 [0.83 - 1.59], p = 0.386
mannonate*	1.16 [0.84 - 1.56], p = 0.336	1.67 [1.21 - 2.29], p = 0.002	1.33 [0.99 - 1.74], p = 0.046	1.84 [1.37 - 2.47], p = <0.001

mannose	1.23 [0.86 - 1.82], p = 0.280	1.06 [0.74 - 1.61], p = 0.751	1.6 [1.1 - 2.4], p = 0.018	1.26 [0.9 - 1.85], p = 0.203
metabolonic lactone sulfate	0.9 [0.63 - 1.27], p = 0.542	0.72 [0.48 - 1.05], p = 0.092	1.54 [1.11 - 2.14], p = 0.010	0.72 [0.51 - 1], p = 0.054
N-stearoyl-sphingosine (d18:1/18:0)*	1.2 [0.85 - 1.7], p = 0.293	1.52 [1.04 - 2.23], p = 0.031	1.41 [1.01 - 1.98], p = 0.042	2.09 [1.48 - 2.98], p = <0.001
pregnenetriol disulfate*	0.94 [0.65 - 1.35], p = 0.721	0.97 [0.67 - 1.45], p = 0.895	1.82 [1.28 - 2.63], p = 0.001	1.3 [0.92 - 1.85], p = 0.136
sphingomyelin (d17:2/16:0, d18:2/15:0)*	1.07 [0.75 - 1.53], p = 0.721	0.96 [0.66 - 1.43], p = 0.845	0.82 [0.6 - 1.15], p = 0.241	0.73 [0.53 - 1.02], p = 0.059
sphingomyelin (d18:0/18:0, d19:0/17:0)*	1.23 [0.86 - 1.73], p = 0.248	1.14 [0.79 - 1.65], p = 0.483	1.3 [0.93 - 1.82], p = 0.123	1.83 [1.3 - 2.62], p = <0.001
sphingomyelin (d18:1/18:1, d18:2/18:0)	1.5 [1.04 - 2.17], p = 0.031	1.94 [1.28 - 3], p = 0.002	0.72 [0.49 - 1.03], p = 0.077	1.4 [0.98 - 2.02], p = 0.070
sphingomyelin (d18:2/14:0, d18:1/14:1)*	1.15 [0.8 - 1.64], p = 0.452	0.95 [0.65 - 1.39], p = 0.793	0.68 [0.48 - 0.97], p = 0.035	0.43 [0.3 - 0.61], p = <0.001
sphingomyelin (d18:2/16:0, d18:1/16:1)*	1.2 [0.86 - 1.69], p = 0.288	1.26 [0.88 - 1.85], p = 0.220	0.68 [0.48 - 0.96], p = 0.028	0.89 [0.65 - 1.25], p = 0.499
sphingomyelin (d18:2/18:1)*	1.43 [1 - 2.03], p = 0.047	1.16 [0.8 - 1.71], p = 0.448	0.65 [0.46 - 0.92], p = 0.014	1.06 [0.76 - 1.5], p = 0.753
sphingomyelin (d18:2/21:0, d16:2/23:0)*	1.16 [0.83 - 1.63], p = 0.384	1.11 [0.77 - 1.61], p = 0.581	0.64 [0.45 - 0.89], p = 0.009	0.75 [0.55 - 1.04], p = 0.081
tartronate (hydroxymalonnate)	0.98 [0.71 - 1.39], p = 0.882	0.76 [0.55 - 1.06], p = 0.094	0.93 [0.67 - 1.34], p = 0.679	0.65 [0.48 - 0.88], p = 0.005

Supplementary Table 6. Associations between maternal BMI-associated metabolites and gestational diabetes or preeclampsia in VDAART.

Odds ratios and 95% confidence intervals from multivariable logistic regression models for each of the 46 BMI-associated maternal serum metabolites in relation to gestational diabetes and preeclampsia at 10–18 weeks and 32–38 weeks of gestation. Models were adjusted for social circumstances (the first principal component of household income, maternal education level, and maternal age at birth), child sex, smoking during pregnancy and self-reported race.

Significant associations ($p < 0.05$) are highlighted in bold in the table.

Reviewer Comment #28

- Lines 135 – 146: These results are in line with expectations and the very probable inferred conclusion is aptly captured by the authors in Lines 143 -146. The authors may consider shortening this paragraph altogether and just refer to Table S5 to arrive at the conclusion. This would create some space to elaborate more on the GDM & PE results.

Authors Response #28

We appreciate the reviewer's suggestion. In response, we have shortened this paragraph to improve clarity and allow greater focus on the gestational diabetes and preeclampsia results.

Specifically, we now summarise the delivery-related outcomes more concisely and direct readers to Table 2 for the full set of results.

Line 155: *"Associations between metabolite scores and delivery-related outcomes were less consistent. Metabolite scores were not significantly associated with caesarean section, though trends were observed for both early and late pregnancy. Induction of birth was associated with the late-pregnancy score ($p = 0.048$) and showed a borderline association with the early-pregnancy score ($p = 0.062$). No associations were found for maternal antibiotics at birth (Table 2). The lack of consistent associations with caesarean section, induction of birth, or maternal antibiotics in our external validation may indicate that the inferences observed in COPSAC2010 might be more related to the direct mechanical consequences of high BMI rather than the metabolic perturbations disturbances associated with BMI."*

Reviewer Comment #29

- Line 147: Restrict subtitle scope to: Independence of maternal metabolite scores associations with pregnancy complications from maternal pre-pregnancy BMI

Authors Response #29

We have complied with this request, and moved our previous subtitle down to the next paragraph where it fits better with the proceeding text. ("**Mediating Role and Independent Predictive Value of Metabolite Scores in Pregnancy Complications**").

We have taken the liberty to edit your suggestion slightly to:

Line 163:

Subtitle: "Independence of the Maternal BMI Associated Metabolite Scores Associations with Pregnancy Complications"

Reviewer Comment #30

- Lines 149 – 151: To improve readability, consider rephrasing as follows: “For both COPSAC2010 and VDAART, we assessed whether,,,” and “When adjusting the maternal metabolite BMI scores for maternal BMI...”

Authors Response #30

We thank the reviewer for this observation. We have updated the manuscript text to clarify that the analysis assessing independent predictive value over BMI was conducted in VDAART only, we have therefore based on your suggestions updated the text to:

Line 165: “We next assessed whether the metabolite scores had independent predictive value over anthropometrically measured BMI in VDAART (Table S7). When adjusting the maternal metabolite BMI scores for maternal BMI, the late pregnancy metabolite score demonstrated some clear distinctions in predictive capacity compared to early pregnancy.”

Reviewer Comment #31

- Lines 152 – 159 – it is observed that in Table S5 that the odds ratios (?; add the statistic information in caption Table S5: e.g. odds ratio per 1 SD change, or whatever the reported statistic is) for COPSAC2010 for both GDM and PE fall in-between VDAART 10-18 and VDAART 23-38 values. This may be indicative for a time to diagnosis association or gradual exacerbation of metabolic dysregulation.

Authors Response #31

We thank the reviewer for this thoughtful observation. While the pattern noted is indeed suggestive of a possible time-dependent effect or progressive metabolic dysregulation during pregnancy, we would caution against direct comparisons of effect sizes between COPSAC2010 and VDAART in this context. The supervised model was trained in COPSAC2010, using maternal BMI as the response variable. Therefore, entering BMI-derived scores back into the same cohort introduces inherent collinearity with the outcome, which may inflate associations and limit interpretability of cross-cohort comparisons.

For this reason, we considered the cleaner inference strategy to be comparing early versus late pregnancy timepoints within the VDAART cohort, where the model was externally applied and not involved in training. Accordingly, we have already included relevant discussion text highlighting the importance of gestational timing, such as:

Line 209: “Furthermore, our results indicate stronger predictive power in late gestation compared to early gestation, highlighting the importance of gestational timing in metabolic risk assessments.”

Line 218: *“Many of our findings align with existing literature, which suggests that metabolomic profiles in pregnancy may enhance predictive capacity for pregnancy outcomes (7,8,12), particularly in late pregnancy compared to early pregnancy. This complements our observation in the VDAART cohort where BMI-metabolite scores showed stronger inference in later pregnancy than in early pregnancy.”*

Additionally, we have updated the caption of Table S7 to clarify that the estimates represent odds ratios per 1 SD change in the metabolite score.

Reviewer Comment #32

-Line 160: add an extra subtitle to introduce the mediation analysis [or effect modification analysis]

Authors Response #32

Thank you, we have addressed this point above in **Authors Response number #29**

Reviewer Comment #33

- Lines 171 – 182: -The explanation regarding incorporation of the both the 16 metabolite score and the 46 metabolite score in the modelling etc is rather confusing and it is interpretation lack transparency. - Provision of the formula for the respective 16 and 46 metabolite scores (in supplementary) and how they were combined in the mutually adjusted regression model may assist the reader. Other than p-values, no comparator OR are provided for the mutually adjusted model to support the claim of significant improvement in predictive capacity. - On a separate note: Given that the 16 metabolites are integral to the 46 metabolites, the reviewer struggles to consider the combination of the 2 scores into a single model justifiable as they are highly correlated. Or, are the authors referring to the use of either metabolite score with maternal pre-pregnancy BMI when they talk about the mutually adjusted model; this may make more sense and appears to be implied in line 182.

Authors Response #33

We thank the reviewer for highlighting this point and agree that clarification is warranted. In our analysis, we compared the full 46-metabolite BMI score with the subset 16-metabolite score (identified through backward elimination mediation analysis) by including both scores in the same multivariable logistic regression model. This was done to evaluate whether the subset score provided *additional predictive value* beyond the full score. While we acknowledge the scores are partially colinear (since the 16 metabolites are a subset of the 46), we found that mutual adjustment was still informative to evaluate which score retained predictive strength when both were modelled together.

To compare the models statistically, we used likelihood ratio tests in R, comparing the base model (with the full 46-metabolite score only) to the mutually adjusted model (including both scores). The statistically significant p-values reported (0.009 for early pregnancy, 0.016 for late pregnancy) reflect improvement in model fit. While we opted not to report odds ratios from the mutually adjusted models due to expected instability from collinearity, we have now clarified this choice in the methods under `Statistical Analysis`

Line 522: *“To formally compare model performance between the full maternal BMI associated metabolite score and the subset of 16 mediating metabolites, we used a likelihood ratio test based on the chi-squared distribution to assess whether adding the subset score significantly improved model fit.”*

We also clarify that Supplementary Table S3 now includes all 46 metabolites used in the full BMI score, along with their component loadings and an indicator of whether each metabolite significantly mediated the association between maternal BMI and gestational diabetes. The 16 mediating metabolites, identified via backward elimination and used to construct the subset score, are highlighted in this table for full transparency.

Finally, we have made the R code used to derive both the 46- and 16-metabolite scores available via GitHub as part of our publication materials, to ensure reproducibility.

Reviewer Comment #34

- general question: Can the authors add a comment on whether the Vitamin D supplementation as being trialed in the VDAART cohort is expected to have an impact on the metabolome profiles of the Vitamin D exposure group.

Authors Response #34

Thank you for this insightful question. We examined the potential influence of the Vitamin D intervention in VDAART on the 32-week metabolite score. In an univariate linear regression model (metabolite_score ~ intervention), there was no association observed ($\beta = 0.01$, $p = 0.848$).

To further assess whether the intervention could confound our findings, we additionally adjusted our outcome models for the Vitamin D intervention. As shown in the table below, the effect estimates for pregnancy complications remained virtually unchanged after adjustment, indicating that the intervention did not materially influence our results.

VDAART Pregnancy Complications	Maternal BMI 32 Week Metabolite Score	32 Week Metabolite Score (Vit D Intervention)
Gestational Diabetes	*** 2.1 [1.48 - 3.03] ($p = <0.001$)	2.1 [1.48 - 3.04] ($p = <0.001$)
Preeclampsia	*** 1.82 [1.23 - 2.72] ($p = 0.003$)	1.81 [1.23 - 2.71] ($p = 0.003$)
Cesarean Section	1.15 [0.98 - 1.35] ($p = 0.095$)	1.15 [0.98 - 1.35] ($p = 0.095$)
Acute	1.11 [0.86 - 1.44] ($p = 0.410$)	1.11 [0.86 - 1.44] ($p = 0.411$)
Elective	1.13 [0.94 - 1.36] ($p = 0.192$)	1.13 [0.94 - 1.36] ($p = 0.191$)
Induction of Birth	1.21 [1 - 1.47] ($p = 0.048$)	1.21 [1 - 1.47] ($p = 0.047$)
Antibiotics Administered to Mother at Birth	1.01 [0.87 - 1.17] ($p = 0.917$)	1.01 [0.87 - 1.17] ($p = 0.905$)

Given the consistency of these results, we decided not to include this additional adjustment in the main manuscript to maintain clarity and focus. Our rationale is that the Vitamin D intervention did not demonstrate a significant association with the metabolite score or materially affect the outcome associations. However, we appreciate the reviewer's suggestion and are happy to provide these results here to reassure regarding potential confounding.

Reviewer Comment #35

- lines 187 – 188: "...we identified specific metabolic disturbances linked to higher maternal pre-pregnancy BMI that may contribute to adverse pregnancy outcomes". Do the authors mean that the metabolite scores of women experiencing adverse pregnancy outcomes are similar to the metabolic scores of women not experiencing adverse pregnancy outcomes but having a higher pre-pregnancy BMI? In other words, women for whom the metabolite based BMI prediction is higher than their actual pre-pregnancy BMIs are found to have an increased risk for GDM or PE? Can the authors clarify their statement accordingly.

Authors Response #35

Thank you for the request to clarify our intended meaning. Our statement refers to the fact that the metabolite score we derived, trained to predict maternal pre-pregnancy BMI, also captured metabolic features that were strongly associated with the risk of pregnancy complications. Specifically, women with a higher metabolite-based BMI score, regardless of their actual BMI (in VDAART, the validation cohort), were at increased risk of gestational diabetes and preeclampsia.

In other words, our findings suggest that a metabolic profile resembling that of higher maternal BMI may predispose to adverse outcomes. Given that our metabolite-based BMI score was derived using a linear modeling approach, these results imply that the metabolomic risk signature associated with higher BMI is not confined to clinically obese individuals but may exert a graded effect across the entire BMI spectrum. Thus, individuals with a normal or borderline BMI but a disproportionately elevated metabolite score may share underlying metabolic disturbances with higher-risk groups, thereby facing increased susceptibility to complications such as gestational diabetes or preeclampsia.

Reviewer Comment #36

- Lines 225 – 238: The authors may want to consider linking this section more explicitly to the conceptual framework that -in general- women with higher BMI have a higher a priori risk already and are therefore more susceptible to any extra metabolic challenge. Inability to cope with the extra metabolic challenge of pregnancy is apparent from a deviation of the maternal BMI – metabolite score correlation; the authors may want to consider referring to the following literature; DOI: [10.1097/00001703-200312000-00002](https://doi.org/10.1097/00001703-200312000-00002)

Authors Response #36

Thank you for this thoughtful suggestion. We agree that pregnancy can act as a physiological stress test, unmasking subclinical metabolic vulnerabilities, as highlighted in

the work by Williams (Curr Opin Obstet Gynecol. 2003). In line with this conceptual framework, our findings suggest that women with higher baseline BMI are already metabolically burdened and thus more susceptible to pregnancy complications. This suggests that a deviation from the expected correlation between anthropometric and metabolite-derived BMI may serve as a proxy for impaired metabolic adaptability. We have included the following text in our discussion to reflect this:

Line 263: *“This suggests that the metabolic alterations of obesity and pregnancy disorders share a common origin and supports our observation that higher baseline BMI levels in VDAART enhance the predictive power of metabolite scores, even in early gestation. This interpretation is in line with the concept of pregnancy as a physiological stress test, which may unmask underlying metabolic vulnerability in women predisposed to cardiometabolic disease later in life (22).”*

Reviewer Comment #37

Line 263: in the context of this manuscript it would be more appropriate to thank the pregnant women who participated in the respective studies rather than their children.

Authors Response #37

Thank you for this relevant perspective

We have accordingly updated our Acknowledgements text to

Line 298: “We express our deepest gratitude to the families of the COPSAC2010 and VDAART cohorts for all their support and commitment.”

Reviewer Comment #38

- General comment: there is a lack of detail provided for all data analysis steps performed like; imputation of missing values (metabolite signals and covariates), scaling, statistical analysis. In particular the various multivariable modelling performed require more technical detail to allow their evaluation in terms of assumptions made, analyst decisions applied.

Authors Response #38

We appreciate the reviewer’s emphasis on transparency and reproducibility in data processing and modeling decisions. While many of these elements are already described in the Methods section of the manuscript, we thank the reviewer for the opportunity to highlight and clarify key technical aspects and the rationale underlying our analytical workflow:

Line 452: *“Data preprocessing involved excluding metabolites with more than 33% missing values. For the remaining missing data, random forest imputation was applied using the missForest R package (v1.5) (29). This method does not assume data are missing at random and has been shown to outperform minimum value imputation in preserving the structure and variability of metabolomics data (30). Before analysis, the metabolome data was log-transformed, centred, and scaled. The dataset for mothers at mid-pregnancy (24 weeks gestation) included a total of 760 annotated metabolites for analysis.”*

This preprocessing pipeline was designed to retain high-quality features while minimising noise from incomplete measurements. The 33% missingness threshold follows commonly accepted standards to ensure stability in downstream imputation. We applied nonparametric random forest imputation to capture complex relationships between metabolites. Log-transformation reduced right-skew, and z-score standardization (centering and scaling) facilitated interpretability across variables and cohorts.

Regarding covariates and imputation:

Line 480: *“Missing covariate data was imputed, based on the available covariate information using the imputePCA function with one component from the missMDA (v1.18) R package.”*

We used imputePCA separately in each cohort to preserve cohort-specific correlation structures among covariates, and to maximise the number of participants for the analysis (provided exposure and outcome data were available). One principal component was retained to model the dominant covariate structure while limiting overfitting.

Regarding metabolite score prediction and scaling:

Line 455: *“Before analysis, the metabolome data was log-transformed, centred, and scaled.”*

This external prediction approach avoided data leakage and ensured independence between training and testing datasets. Scores were z-scaled so odds ratios for pregnancy complications reflect per standard deviation (SD) change, as noted in the Statistical Analysis section.

Regarding our multivariable modeling framework and assumptions:

The manuscript describes our use of multivariable logistic regression models to evaluate the associations between maternal pre-pregnancy BMI, BMI-associated metabolite scores, and pregnancy complications. As stated in the Statistical Analysis section:

Line 484: *“Multivariable logistic regression models were used to determine the associations of maternal pre-pregnancy BMI, and associated metabolite scores on pregnancy complications... All multivariable models were adjusted for the previously mentioned covariates.”*

To address the reviewer’s request for additional detail, we have now included the following statement under the Covariates section of the Methods:

Line 470: *“We included covariates based on a directed acyclic graph (DAG) constructed to guide covariate selection using causal reasoning (31).”*

This approach ensured principled selection of confounders relevant to both exposure and outcome while avoiding adjustment for potential mediators or colliders. The selected covariates were applied consistently across cohorts and models.

Reviewer Comment #39

- line 438: what is meant with twin pairs: are these technical replicates? If yes, please adopt this terminology.

Authors Response #39

Thank you for this question. The term twin pairs refers to biological twins (i.e., children from the same pregnancy). Our dataset is structured at the child level, so twin pregnancies are represented twice. Since this study focuses on maternal exposures and pregnancy outcomes, we excluded one child per twin pair to ensure independence of observations and avoid duplicate pregnancy records. We have clarified this in the manuscript.

Line 491: “As the data is structured at the child level, we excluded one child from each twin pair to avoid duplicate pregnancy records and ensure independent observations.”

Reviewer Comment #40

- lines 446 – 457: - The authors should add more detail regarding methodology used (references), assumptions, models used, etc. Currently It is unclear how the backward elimination was performed and assessed. Is the backward elimination applied to the second multivariable model and assessed in the first model where the metabolite score is the dependent variable?

Authors Response #40

We thank the reviewer for this valuable comment. To address this, we have now updated the Methods section to provide greater clarity on the backward elimination process and the associated modeling framework. Specifically, we have:

Clarified that backward elimination was applied to the first model, the composite metabolite score predicting maternal BMI, and that the mediating effect was assessed using the second model where this metabolite score was included as the mediator of the relationship between maternal BMI and the outcome.

Specified that the elimination procedure was guided by the magnitude of the mediation effect (average causal mediation effect, ACME) obtained from formal causal mediation analysis.

Included a reference to our prior work in Nature Metabolism (doi:10.1038/s41591-022-01765-2), where the same backward elimination framework was employed.

Provided the R code used for the elimination loop and mediation analysis, made available for full transparency.

To summarise the methodology in brief:

We used a pair of multivariable models:

1. A linear regression model regressing the composite metabolite score on maternal BMI and covariates.
2. A logistic regression model regressing the pregnancy complication on maternal BMI, the composite score, and the same covariates.

Mediation was then assessed using the mediate function from the mediation R package, estimating the ACME. We applied a backward elimination loop, sequentially removing one metabolite at a time from the metabolite score and recomputing the mediation. At each iteration, the metabolite whose removal *maximally increased* the ACME was dropped, continuing until no further improvement was gained. This approach aimed to identify a metabolite signature that retained, or strengthened, the observed mediation effect.

To aid reproducibility, we have updated our description of this methodology in the Methods section:

Line 506: *“To elucidate the role of metabolites as potential biomarkers or mediators in the association between maternal pre-pregnancy BMI and pregnancy complications, we employed a systematic backward elimination strategy (33). We prioritised outcomes for mediation analysis where both maternal pre-pregnancy BMI and BMI metabolite scores exhibited significant associations in both cohorts. This approach ensures that the mediation analysis focuses on outcomes where metabolomic factors may play a direct role in mediating the relationship between maternal BMI and pregnancy complications, thus providing insights into potential biological mechanisms. Our analysis approach was designed to iteratively exclude metabolites, aiming to identify those with the most profound mediating influence on the outcome. We used two multivariable models: one linear regression model linking a composite metabolite score to maternal pre-pregnancy BMI and covariates, and another logistic regression model assessing its mediating role between maternal pre-pregnancy BMI and pregnancy complication outcomes, using the same covariate structure. Backward elimination was applied to the first model by iteratively removing individual metabolites from the composite score and recalculating the mediation effect. At each step, the metabolite whose removal most increased the average causal mediation effect (ACME) was dropped, and the process continued until no further gain was observed. This causal mediation analysis was conducted with the mediation package in R (v4.5.0), with 10,000 simulations performed at each iteration. To formally compare model performance between the full maternal BMI associated metabolite score and the subset of 16 mediating metabolites, we used a likelihood ratio test based on the chi-squared distribution to assess whether adding the subset score significantly improved model fit.”*

We trust this clarifies the approach and strengthens the methodological transparency of the study.

Reviewer Comment #41

- Lines 452 – 457: The rationale to only consider mediation analysis for the outcome GDM is better placed at beginning of the paragraph.

Authors Response #41

We agree with this comment and have placed this towards the beginning of the paragraph (see text above, in previous Authors Response #40).

Reviewer Comment #42

Lines 460 – 461: annotate the use case for each of the listed R packages.

Authors Response #42

Thank you for this suggestion, we have updated the text accordingly:

Line 528: “Other R packages utilised in this analysis include tidyverse (v1.3.1) for general data processing and visualisation, dplyr (v1.0.10) for data wrangling, broom (v0.7.12) for tidying model outputs, lubridate (v1.8.0) for handling date variables, ggpubr (v0.4.0) for producing publication-ready plots, and tableone (v0.13.0) for generating descriptive baseline tables.”

—

Finally, the authors would like to express our sincere thanks for the time and effort you invested in reviewing our manuscript. We greatly appreciate your thoughtful and constructive feedback, which has helped us improve the clarity, transparency, and robustness of our work. We hope the revisions outlined above satisfactorily address your comments and strengthen the manuscript.

Point-by-point Reviewer Response to:

A Metabolomic Signature of Maternal BMI is Associated with Pregnancy Complications: Insights from the COPSAC2010 and VDAART Mother-Child Cohorts

David Horner, MD PhD, Rebecca Vinding, MD PhD, Tingting Wang, PhD, Mina Ali, PhD, Mario Lovric, PhD, Nicole Prince, PhD, Jessica Lasky-Su, ScD, Klaus Bønnelykke, MD PhD, Jakob Stokholm, MD PhD, Bo Chawes, MD PhD DMSc, Morten Arendt Rasmussen, PhD.

Horner et al. engaged constructively with the reviewers' comments and questions and made changes to their manuscript accordingly. The data presentation and general outlay of the manuscript markedly improved. The authors are commended for taking on the reviewers' critiques with rigor. Yet, some deficiencies remain: in the first review iteration, the reviewer did not spot the absence of correction for multiple testing when evaluating BMI and metabolite score associations with the multiple pregnancy outcomes, increasing the likelihood of false positive findings. No justification for not applying multiple testing correction was given. Some further recommendations / comments are therefore provided below.

Thank you once again for your thorough review of our work. We appreciate your comments and believe our manuscript has been improved as a result.

We thank you as well for your comments regarding our active engagement with the review process. Further, we appreciate the reviewer's concern about multiple testing.

Prior to answering your specific points below, we would like to address the reviewers concern regarding multiple testing which appear in numerous responses. My co-author and I have considered this carefully and respectfully disagree that a formal false discovery rate (FDR) or Bonferroni correction is appropriate for our study. Our reasoning is threefold:

- 1. Our study is hypothesis-driven and we use pre-specified outcomes**

The five pregnancy complications (gestational diabetes, preeclampsia, cesarean section, induction, and intrapartum antibiotics) were chosen *a priori* based on strong biological plausibility and prior evidence linking maternal BMI with these outcomes. This is therefore not an exploratory "fishing expedition" across numerous outcomes, but a targeted test of a clearly motivated hypothesis. In such contexts, applying stringent multiplicity corrections risks inflating type II error and obscuring biologically meaningful associations. To reinforce this point, we have revised our Introduction to state the underlying hypothesis explicitly: namely, that BMI-associated metabolic perturbations mediate the established link between maternal BMI and pregnancy complications.

Line 73: “On this basis, we hypothesise that BMI-associated metabolic perturbations mediate the link between maternal BMI and pregnancy complications, and that identifying these perturbations can provide mechanistic insight beyond anthropometric measures.”

2. **Interdependence of outcomes**

Several of these endpoints are correlated both statistically and biologically. For example, gestational diabetes is a known risk factor for preeclampsia, and both often manifest in preterm or complicated deliveries. Likewise, cesarean section, induction, and intrapartum antibiotics are interrelated birth interventions. Standard multiplicity corrections assume independent tests, which is not the case here. Adjusting as though they were independent comparisons would therefore be overly conservative.

3. **Replication and transparency**

To mitigate risk of false discoveries, we validated our findings in a fully independent, large mother child cohort (VDAART). Replication across cohorts offers a stronger safeguard against spurious associations than a purely statistical correction. In addition, we present all exact p-values and 95% confidence intervals, allowing readers to judge the strength and precision of associations transparently, comparing findings and inference between the training, and testing cohort.

This approach is consistent with methodological guidance in clinical epidemiology, which cautions against automatic correction when testing a small, biologically coherent set of pre-specified outcomes. Rothman has argued that “no adjustments are needed for multiple comparisons” when results are interpreted in context (PMID 2081237). Similar positions are supported by Perneger (PMID 9553006), Bender & Lange (PMID 11297884), and Feise (PMID 12069695). These references collectively highlight that multiplicity corrections can be counterproductive in hypothesis-driven studies with a small number of correlated outcomes.

For exploratory analyses (e.g., the single-metabolite results in Table S6 added in our previous revision), we have clarified in the Table legend that these are hypothesis-generating and that no multiple testing correction was applied.

Reviewer #2:

Reviewer Comment #43

Reviewer comment #17: Reviewer accepts the authors’ modelling choice rationalisation; it is noted that the equivalent of Figure S4 for VDAART 32 – 38 wks is not generated. Given that prediction of the metabolite panel at this time point is most performant, the authors may want to add this Figure as well.

Authors Response #43

Thank you for your comment. We would highlight that we have already added the figure you suggest in our previous revision, based on your previous comments (Authors response #26). This is incorporated in our manuscript as Supplementary Figure 4.

Reviewer Comment #44

Reviewer comment #20: The authors are thanked for provision of the missingness list. The original question remains: is there a pattern in missingness for the 6 xenobiotic metabolites of the 46 panels across the outcomes.

Authors Response #44

We thank the reviewer for clarifying their question. Our primary focus was on the robustness of the metabolite-predicted BMI model rather than pathway-specific differences in missingness. As such, we did not stratify missingness analyses by individual super- or sub-pathways (e.g., xenobiotics), and this was not a primary focus of our study. However, to address the reviewer's request, we have provided the missingness values for the subset of metabolites classified under the "Xenobiotics" super-pathway (see Reviewer file attached to our submission). As expected, xenobiotics as a whole were among the pathways with the highest proportion of missing values, reflecting their exogenous origin (e.g., diet, medications etc). Of relevance to our work, of the 46 metabolites included in our BMI-associated metabolite set, only six were xenobiotics (3-phenylpropionate (hydrocinnamate), tartronate (hydroxymalonate), 2,6-dihydroxybenzoic acid, mannonate, 4-ethylphenyl sulfate, and ergothioneine), all of which showed minimal or no missingness (0–3.3%). This low missingness may not represent a biological pattern per se, but rather the result of our exclusion criteria, where metabolites with >33% missingness were removed prior to imputation and modeling. Of relevance, these retained xenobiotics are those consistently detected across individuals and trained in our models to reflect higher maternal BMI.

To further support potential patterns of missingness in metabolites, we refer the reviewer to our previous analysis of missingness presented in the manuscript (Table S4). There, we evaluated whether missingness of the selected 46 BMI-associated metabolites (including the xenobiotics) was associated with pregnancy complications.

Reviewer Comment #45

Reviewer comment #28: Given that there was no correction applied for multiple testing, the reviewer would still advise not to spend any discussion space on the association between the metabolite scores with the delivery-associated outcomes other than the conclusion. It is expected that upon correcting for multiple testing, no significant associations will remain in COPSAC2010 as well.

Authors Response #45

We appreciate the reviewer's advice. As suggested in the previous round, we have already shortened the delivery-related outcomes paragraph to a more concise summary and now refer readers directly to Table 2 for the full set of results. We believe this balances the space allocated to each outcome, while ensuring that gestational diabetes and preeclampsia, the most robust and biologically grounded findings, remain the main focus of the text. At the

same time, we consider it important to retain a brief mention of the delivery-related outcomes in the results narrative, as providing inference may help the reader understand the basis for our conclusion that these outcomes likely reflect direct mechanical consequences of high BMI rather than BMI-associated metabolic perturbations.

Line 298: *“Maternal BMI metabolite scores in VDAART provided nuanced insights into pregnancy complications at different gestational stages. Early-pregnancy metabolite scores (10-18 weeks) demonstrated weaker associations with gestational diabetes (OR 1.51 [1.08 - 2.14], $p = 0.018$) compared to late-pregnancy scores (32-38 weeks) (OR 2.10 [1.48 - 3.03], $p < 0.001$). Similarly, late-pregnancy metabolite scores were significantly associated with preeclampsia (OR 1.82 [1.23 - 2.72], $p = 0.003$), while early-pregnancy scores were not significantly associated with preeclampsia (OR 1.40 [0.99 - 2.00], $p = 0.060$).*

Associations between metabolite scores and delivery-related outcomes were less consistent. Metabolite scores were not significantly associated with caesarean section, though trends were observed for both early and late pregnancy. Induction of birth was associated with the late-pregnancy score ($p = 0.048$) and showed a borderline association with the early-pregnancy score ($p = 0.062$). No associations were found for maternal antibiotics at birth (Table 2). The lack of consistent associations with caesarean section, induction of birth, or maternal antibiotics in our external validation may indicate that the inferences observed in COPSAC2010 might be more related to the direct mechanical consequences of high BMI rather than the metabolic perturbations disturbances associated with BMI.”

With respect to multiple testing, applying formal FDR or Bonferroni correction across such a small, correlated set of outcomes would be overly conservative and risk type II error, a concern we elaborate on in prior detailed response to the reviewer’s broader comments on multiple testing. To ensure transparency, we present exact p-values and confidence intervals, and we rely on replication in the independent VDAART cohort as the most stringent test of reproducibility.

Reviewer Comment #46

a. Lines 59-60: Suggested rephrasing - “We also performed mediation analysis for these scenarios where both BMI and BMI-associated metabolite scores associate with adverse pregnancy outcomes to elicit specific metabolites which likely mediate the effect between maternal BMI and adverse pregnancy outcomes.”

b. Line 61: Replace “contrasting” by “evaluating”

c. Line 63: extend the sentence as follows “...for predicting maternal health outcomes across gestation.”

Authors Response #46

We thank the reviewer for this helpful suggestion. We have revised the sentence to improve clarity and precision. The text now reads:

Line 79: “We performed mediation analyses where both BMI and BMI-related metabolite scores showed associations with adverse pregnancy outcomes, to identify metabolites potentially mediating these effects.”

We have also replaced “contrasting” with “evaluating” as suggested.

We have extended the sentence by adding “across gestation” as suggested.

Reviewer Comment #47

a. Line 78: the number of metabolites of 640 considered for analysis is based on their common presence in the COPSAC2010 and VDAART datasets, this is not apparent from the language in the results; it is only mentioned in the methods. Consider rephrasing as follows “...46 metabolites out of 640 metabolites available for modelling...”

Authors Response #47

Thank you for your suggestion, we have made this change.

Reviewer Comment #48

B. Lines 86-92; Consider removal of section discussing assumed non-randomness of missingness altogether (remove Table S4 as well; if retained, the heading of column 2 should be corrected (remove “p-value”). When correcting for multiple testing (at least 5 comparisons are made: GDM, PE, C-section, induction, antibiotics), the observed association will not reach significance. Here and in all other outcome analyses, the authors should correct for multiple testing.

c. Lines 95 -96: Update results upon application of multiple testing correction; it is expected that several of the associations will be rendered non-significant, simplifying interpretation of the results.

d. Lines 104 -113: Update results upon application of multiple testing correction; it is expected that several of the associations will be rendered non-significant, simplifying interpretation of the results.

f. Lines 127 – 133: Update results upon application of multiple testing correction

k. Lines 149 – 154: Update results upon application of multiple testing correction; same conclusions expected

l. Lines 165 – 175: Update results upon application of multiple testing correction; same conclusions expected

Authors Response #48

We thank the reviewer for these comments. As outlined above, we respectfully disagree that formal multiple testing correction is appropriate for this study. Our analysis focuses on a small number of pre-specified, biologically interrelated pregnancy outcomes, where multiplicity adjustments would be overly conservative and risk obscuring meaningful

associations. Instead, we provide exact p-values and 95% confidence intervals, and we validate findings in an independent cohort (VDAART), which we believe offers a stronger safeguard against false positives. Accordingly, we have not applied FDR or Bonferroni correction in the specific sections referenced (lines 86–92, 95–96, 104–113, 149–154, and 165–175), but we have retained clear reporting of results and believe our conclusions are reasoned and appropriately weighted given the quality of the data and the analytical approach used.

Reviewer Comment #49

e. Line 119: correct ..."higher rates of maternal smoking..." to "lower rates of maternal smoking..."

g. Line 135: consider rephrasing as follows: "...prediction scores to metabolomics data obtained from maternal blood samples..."

Authors Response #49

Thank you for your comments.

We have changed "higher rates of maternal smoking" to "lower rates of maternal smoking" as suggested (and we appreciate your efforts for identifying this error).

To your suggested change for line 135 we have updated the text accordingly based on your suggestion:

"We applied the 46-metabolite COPSAC2010-trained prediction scores to metabolomics data obtained from maternal blood samples in VDAART at the gestational windows: early (10–18 weeks) and late pregnancy (32–38 weeks)."

Reviewer Comment #50

h. Line 139: Clarify figure S4 is plotting the BMI-associated metabolite vs BMI for the 10-18 weeks gestation. Consider adding an additional figure plotting the data for the 32- 38 weeks gestation as well.

Authors Response #50

We thank the reviewer for this helpful suggestion and highlighting the oversight that we did not include this was the early gestational timepoint. We have now clarified in both the main text and figure legend that Figure S4 illustrates the relationship between the maternal BMI metabolite score and measured pre-pregnancy BMI in the VDAART cohort at early gestation (10–18 weeks), with women experiencing pregnancy complications highlighted against the broader cohort. We have decided not to include a similar figure for 32–38 weeks, as this would essentially replicate the early gestation pattern (already an attempted replication) and does not add substantive weight to the manuscript.

Line 294: "Figure S4 illustrates the relationship between the maternal BMI metabolite score and measured pre-pregnancy BMI in the VDAART cohort in early gestation (10-18 weeks), with women experiencing specific pregnancy complications highlighted against the broader cohort."

Figure S4 Text: “**Figure S4.** Relationship between maternal BMI metabolite score and measured pre-pregnancy BMI in the VDAART cohort at early gestation (10–18 weeks). Women experiencing specific pregnancy complications are highlighted against the broader cohort.”

Reviewer Comment #51

i. Line 144: Replace “multivariable modelling” by “logistic regression analysis”

Authors Response #51

Thank you for the suggestion, our intention with our phrasing for our VDAART replication analysis was to highlight these are not univariate analysis (exposure vs outcome) but rather considered potential confounders/covariates. We have also done so when writing about the COPSAC2010 analysis.

Line 250: “Maternal pre-pregnancy BMI was significantly associated with several pregnancy complications in multivariable models adjusted for social circumstances, child sex, maternal smoking, and maternal dietary patterns (Table 2).”

We have decided not to change “multivariable modelling” to “logistic regression analysis” as suggested, as readers can work out these are logistic regressions by the stated odds ratio results, and we believe our methodology is already adequately described in the Methods section (as below):

Line 171: Multivariable logistic regression models were used to determine the associations of maternal pre-pregnancy BMI, and associated metabolite scores on pregnancy complications.

Reviewer Comment #52

j. Line 146-148: 1) Simplify the sentence as follows: “The associations between each of the 46...(Table S6) were also examined. 2) Move sentence to line 150 and put it prior to the sentence “Early pregnancy metabolite scores...” 3) add to caption of Table S6 that the single metabolite association analysis is exploratory and that no multiple testing correction was applied.

Authors Response #52

Thank you for these comments.

We have changed the text accordingly to

Line 299: “The associations between each of the 46 BMI-related metabolites at both pregnancy timepoints and the risk of gestational diabetes and preeclampsia were also examined and can be found in Table S6”. We have also moved the text to before “Early pregnancy metabolite scores...” as suggested.

We have also added the following text to Table S6 following your suggestion: “...of note these analyses are hypothesis-generating and presented without multiple testing correction.”

Reviewer Comment #53

m. Line 168 and Line 172: change “metabolite score” to “BMI-adjusted metabolite score”

n. Line 174: replace “crucial” with “informative”

o. Lines 184-185: rephrase as follows ..., derived from plant sources, may confer some protection from GDM risk, evidenced by...

p. Line 189: replace “analysis” with “cohort”

Authors Response #53

Thank you for your comments. We have made the changes as suggested regarding “BMI-adjusted metabolite score”

We have changed “crucial” to “informative” as suggested.

We have changed to “derived from plant sources, may confer some protection from gestational diabetes risk, evidenced by...” as suggested.

We have changed to “In the validation cohort, we predicted a metabolite score using the subset of mediating metabolites identified in COPSAC2010.” as suggested.

Reviewer Comment #54

q. Lines 193 – 198: addressed in above commentary to reply to reviewer comment #33. Additional question: does the 16 metabolite score also improves preeclampsia prediction?

Authors Response #54

Thank you for the question.

We would highlight that the 16 subset of metabolites are identified via backward elimination mediation analysis trained in COPSAC2010 - against the outcome of gestational diabetes.

As previously reported in our manuscript we identified as significant association of the full BMI metabolite model (46 metabolites) and preeclampsia in the VDAART cohort at the late gestational timepoint, and directionally but not statistically significantly in early pregnancy

Line 303: “Similarly, late-pregnancy metabolite scores were significantly associated with preeclampsia (OR 1.82 [1.23 - 2.72], $p = 0.003$), while early-pregnancy scores were not significantly associated with preeclampsia (OR 1.40 [0.99 - 2.00], $p = 0.060$).”

Having run this extra analysis we can share that using this subset metabolite model, does not improve prediction for preeclampsia over the full model at either the early timepoint (anova comparison $p=0.596$) or at the late gestational timepoint (anova comparison $p = 0.282$).

We note, however, that interpretation of the odds ratios suggests the mediating subset provides a slightly stronger signal for preeclampsia at 32 weeks compared with the early

timepoint. Nonetheless, formal comparison by ANOVA shows that this subset model does not outperform the full 46-metabolite model in predicting preeclampsia at either gestational stage. This indicates that while the subset may hint at relevant biology (despite trained on a separate pregnancy complication), the full metabolite model retains similar predictive value.

Reviewer Comment #55

- a. Line 225: extend sentence as follows: “also closer to any adverse pregnancy outcomes occurring.”
- b. Line 232: replace “cohorts” with “pregnancy populations”
- c. Line 234: Start sentence as follows: In line with our findings, ceramide species....
- d. Line 240: more accurate (?): ... show significant differences between pregnancies with and without gestational diabetes.

Authors Response #55

We have made all the above changes to the discussion text as suggested.

Reviewer Comment #56

Line 245: Consider rephrasing as follows: “The reliability of our findings made in the COPSAC2010 cohort was confirmed through external validation in the VDAART cohort wherein the maternal BMI ...”

Authors Response #56

Thank you for this suggestion. We can confirm we have changed the text to:

Line 397: *“The reliability of our findings made in the COPSAC2010 cohort was confirmed through external validation in the VDAART cohort, wherein the maternal BMI metabolite score and anthropometric BMI were strongly correlated, and our metabolome modelling provided additional benefit in predicting pregnancy complications.”*

Reviewer Comment #57

Lines 247 – 253: this section of the discussion appears in conflict with the results as presented in lines 104-113?

Authors Response #57

Thank you for identifying this section of text. We agree that it was not coherent with the main results and stemmed from an earlier analysis we later decided to remove. We have therefore deleted this text from the discussion, which now reads coherently and consistently with the presented results.

Reviewer Comment #58

Lines 282 – 284: the mentioned comparators are diagnostic criteria. More appropriate risk biomarker comparators are HbA1c for gestational diabetes screening in first trimester or PIGF and s-Flt1/PIGF for respectively early pregnancy and late pregnancy preeclampsia risk screening.s as presented in lines 104-113?

Authors Response #58

Thank you for pointing this out. We agree that the original text was not appropriate, as it conflated benchmarking risk prediction with clinical diagnostic criteria. We have revised the sentence to read:

Line 428: *“While we observed improved predictive capacity with BMI-associated metabolites, we did not benchmark our risk prediction models against established clinical screening tools currently used in practice for gestational diabetes or preeclampsia.”*

Reviewer Comment #59

a. Reference 19: no authors are listed b.

Reference 20: no authors are listed

Authors Response #59

We have updated these reference to ensure they include the authors names.

–

Many thanks again for your comments, which have greatly improved the quality of this manuscript/work.

David Horner, MD, PhD

Point-by-point Reviewer Response to:

A Metabolomic Signature of Maternal BMI is Associated with Pregnancy Complications Across Two Independent Pregnancy Cohorts

David Horner, MD PhD, Rebecca Vinding, MD PhD, Tingting Wang, PhD, Mina Ali, PhD, Mario Lovric, PhD, Nicole Prince, PhD, Jessica Lasky-Su, ScD, Klaus Bønnelykke, MD PhD, Jakob Stokholm, MD PhD, Bo Chawes, MD PhD DMSc, Morten Arendt Rasmussen, PhD.

—

We thank the Editor and Reviewer #2 for the thoughtful, constructive assessment and for recognising that our revisions meet the scientific requirements. We have implemented all requested changes and minor editorial edits. We submit a clean version, a marked-up version with tracked changes, and the completed revision table detailing every change and its manuscript location.

Reviewer #2:

Minor editorial edits

Line 133: Replaced “UPLC(-) MS/MS” and “HILIC/UPLC(-) MS/MS” with “UPLC-ESI(-) MS/MS” and “HILIC UPLC-ESI(-) MS/MS.” **Done.**

Line 158: Added full stop after “(19).” **Done.**

Line 164: Added full stop at sentence end. **Done.**

Line 209: Removed spurious full stop. **Done.**

Line 316: Rephrased to “Next, we assessed...” **Done.**

Line 320: Clarified “at 32-38 weeks...” **Done.**

Line 323: Corrected to “the 32-38 weeks BMI-adjusted...” **Done.**

—

We thank the reviewer and editor for their time and expertise, which we believe has greatly improved our manuscript.

Kind Regards,

David Horner